# Taming Text-to-Sounding Video Generation via Advanced Modality Condition and Interaction

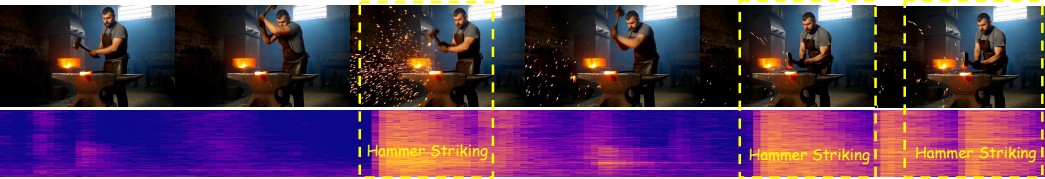

Video Caption: … A strong blacksmith forcefully **strikes a hammer** on glowing iron in the workshop …
Audio Caption: The resounding clang of a heavy **hammer striking metal** …

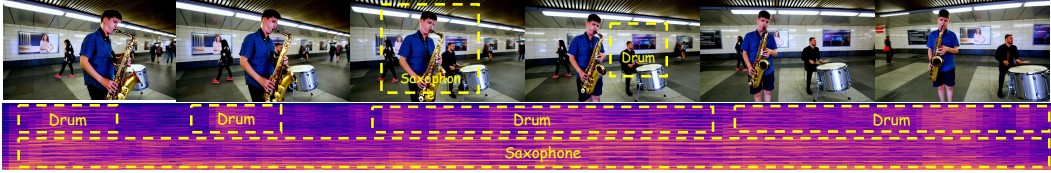

Video Caption: … In a bustling subway station… **A young man** … his golden **saxophone** held close as he plays with fervent passion … Behind him, another musician, **clad in black**, sits with a large **metallic drum**.……
Audio Caption: The rich, soulful notes of a **saxophone**, blending with the rhythmic metallic **drum**.

Figure 1: Examples of sounding videos generated by our BridgeDiT model, showcasing high quality, temporal synchronization, and text alignment. Our method generates high-fidelity video frames and detailed audio spectrograms that remain faithful to the given text prompts. Critically, as highlighted in the dashed boxes, the generated audio and video are precisely synchronized, demonstrating strong temporal coherence between visual events and their corresponding sounds. More cases are shown in the anonymous demo page `https://bridgedit-t2sv.github.io`.

## Abstract

This study focuses on a challenging yet promising task, Text-to-Sounding-Video (T2SV) generation, which aims to generate a video with synchronized audio from text conditions, meanwhile ensuring both modalities are aligned with text. Despite progress in joint audio-video training, two critical challenges still remain unaddressed: (1) a single, shared text caption where the text for video is equal to the text for audio ($T_V = T_A$) often creates modal interference, confusing the pretrained backbones, and (2) the optimal mechanism for cross-modal feature interaction remains unclear. To address these challenges, we first propose the Hierarchical Visual-Grounded Captioning (HVGC) framework that generates pairs of disentangled captions, a video caption ($T_V$), and an audio caption ($T_A$), eliminating interference at the conditioning stage. Based on HVGC, we further introduce BridgeDiT, a novel dual-tower diffusion transformer, which employs a Dual CrossAttention (DCA) mechanism that acts as a robust "bridge" to enable a symmetric, bidirectional exchange of information, achieving both semantic and temporal synchronization. Extensive experiments on three benchmark datasets, supported by human evaluations, demonstrate that our method achieves state-of-the-art results on most metrics. Comprehensive ablation studies further validate the effectiveness of our contributions, offering key insights for the future T2SV task. All the codes and checkpoints will be publicly released.

# 1 INTRODUCTION

Human perception is inherently multi-sensory, with vision and sound tightly coupled. Generating videos with synchronized audio from text (Text-to-Sounding-Video, T2SV) represents a crucial step toward world-modeling. Recent years have witnessed rapid progress in Text-to-Video (T2V) (Blattmann et al., 2023; Brooks et al., 2024; Zheng et al., 2024; Weijie Kong, 2024; Wan et al., 2025) and Text-to-Audio (T2A) (Liu et al., 2023b; Huang et al., 2023; Wang et al., 2025a; Evans et al., 2025) generation. With these unimodal capabilities becoming increasingly mature, the community naturally shifts attention to the more challenging task of T2SV (Tang et al., 2023; Liu et al., 2024a; Ishii et al., 2024; Liu et al., 2025; Weng et al., 2025).

Prior strategies for T2SV suffer from critical limitations. The simple approach of generating video and audio independently with T2V and T2A models fails to achieve temporal synchronization. Pipelined methods (e.g., T→V→A or T→A→V) attempt to address this, but they suffer from error accumulation. This is because the second-stage generative model (Video-to-Audio (Wang et al., 2024b; Cheng et al., 2025b;a; Wang et al., 2025a) or Audio-to-Video (Jeong et al., 2023; Cao et al., 2023; Zhang et al., 2024)), having been trained only on ground-truth data, cannot correct the errors from the first stage and often amplifies them. To overcome these limitations, research has increasingly shifted toward joint video-audio generation, where both modalities are synthesized simultaneously. The single-tower paradigm (Ruan et al., 2023; Tang et al., 2023; Sun et al., 2024; Wang et al., 2024a; Zhao et al., 2025), which learns the audio-video joint distribution from scratch in one shared model, is often data-intensive and complex to optimize, demanding significant computational resources and often struggling with training stability. Thus, the dual-tower architecture (Ishii et al., 2024; Liu et al., 2024a; 2025; Wang et al., 2025b; Weng et al., 2025) has emerged as the dominant approach. This strategy leverages pretrained T2V and T2A backbones and connects them with a lightweight and trainable interaction module, enabling the generation of synchronized sounding videos without the large cost of from-scratch training. Despite its promise, this paradigm still faces two fundamental yet under-explored challenges:

*C1. The Conditioning Problem*: Dual-tower framework is typically initialized with unimodal backbones (T2V, T2A), where each backbone (tower) is pretrained with modality-specific caption, however current dual-tower methods (Liu et al., 2024a; 2025; Zhao et al., 2025; Weng et al., 2025) often use a shared caption for both towers ($T_V = T_A$), mixing visual and auditory conditions. This mixture leads to a modal interference problem: text that is semantically relevant for one modality often appears as irrelevant noise to the other. For example, given the text "a red car emits a sharp honk", the video tower is forced to process the auditory text "sharp honk", while the audio tower is forced to interpret the visual attribute "red". Such modal interference pushes both towers into out-of-distribution text condition, thereby degrading performance.

*C2. The Interaction Problem*: The interaction module is the architectural component responsible for exchanging information between the video and audio towers. However, its optimal design still remains unsolved. The core challenge is enabling an effective yet efficient exchange of features, which is essential for ensuring that the final output is synchronized both semantically and temporally.

In this work, we first address the conditioning problem by introducing Hierarchical Visual-Grounded Captioning (HVGC) framework. HVGC satisfies two critical requirements: (1) provide disentangled, modality-pure text captions for each tower, aligning with their pretraining; and (2) ensure the accuracy of these captions. While direct Audio LLMs (Chu et al., 2024) can provide separate audio captions, they often yield inaccurate or noisy descriptions due to the inherent information sparsity of raw audio, leading to severe hallucinations (Nishimura et al., 2024; Kuan et al., 2024). HVGC rectifies this by employing visual grounding throughout its three-stage pipeline: (i) generating a detailed visual description, (ii) extracting auditory-relevant concepts from it, and (iii) producing a modality-pure audio caption that remains robustly grounded in the visual context. This design ensures both the separation and accuracy crucial for text conditioning. Building on HVGC, we further propose BridgeDiT, a dual-tower architecture with a Dual CrossAttention (DCA) fusion mechanism. This design enables a symmetric, bidirectional information exchange between the video and audio towers. Extensive experiments together with human evaluation demonstrate that our model achieves state-of-the-art results. Furthermore, we conduct detailed ablation studies that validate the critical role of our HVGC framework and, through comparisons with alternative fusion mechanisms, prove the superiority of our Dual Cross-Attention fusion mechanism.

In summary, our main contributions are as followed: (i) a novel Hierarchical Visual-Grounded Captioning (HVGC) framework that generates disentangled text caption to eliminate modal interference in T2SV task; (ii) the BridgeDiT architecture, featuring a Dual CrossAttention (DCA) mechanism for effective and efficient cross-modal fusion; and (iii) comprehensive experiments and analyses that demonstrate state-of-the-art performance and provide valuable insights into caption pipeline and architecture design choice.

## 2 RELATED WORKS

**Text-Condition Single Modal Generation**   Text-condition single modal generation, including Text-to-Video (T2V) (Wang et al., 2023; Zheng et al., 2024; Lin et al., 2024; Weijie Kong, 2024; Kuaishou, 2024; Brooks et al., 2024; Wan et al., 2025) and Text-to-Audio (T2A) (Liu et al., 2023b; 2024b; Evans et al., 2025; Huang et al., 2023), has become a prominent area of research in recent years. Both domains have followed a parallel evolution: architectures have advanced from UN-ets (Özgün Çiçek et al., 2016) to the state-of-the-art Diffusion Transformer (DiT) (Peebles & Xie, 2022), while training paradigms have shifted from DDPM (Ho et al., 2020) to more efficient methods like EDM (Karras et al., 2022) and flow matching (Lipman et al., 2022). While current T2V and T2A models can independently generate high-quality content, they struggle to generate videos with semantically and temporally synchronized sound, which is addressed in this work.

**Video-Audio Cross-Modal Generation**   To improve semantic and temporal synchronization, some works explore audio-video cross-modal generation for T2SV task. This includes Video-to-Audio (V2A) generation (Wang et al., 2024b; Cheng et al., 2025b; Xing et al., 2024; Cheng et al., 2025a; Wang et al., 2025a), which uses video to condition audio generation, and Audio-to-Video (A2V) generation (Lee et al., 2022; Jeong et al., 2023; Cao et al., 2023), which uses audio to condition video generation. These unidirectional models can be chained into pipelines (e.g., T→V→A or T→A→V) to achieve more synchronized audio-visual content than independent generation. However, these pipelined methods suffer from error accumulation (Liu et al., 2024a; 2025) problem. Since these cross-modal models are trained on ground-truth data, any artifacts or inconsistencies from the initial text-conditional stage (Guan et al., 2025) are inevitably propagated, leading to sub-optimal final results. To avoid this error accumulation problem, we instead pursue a joint generation approach where both modalities are created in a single step.

**Text-Condition Audio-Video Joint Generation**   To overcome the error accumulation of pipelined methods, recent research has shifted towards audio-video joint generation. Existing methods largely follow two paradigms: single-tower and dual-tower. The single-tower approach learns the joint audio-video distribution from scratch (Ruan et al., 2023; Tang et al., 2023; Sun et al., 2024; Wang et al., 2024a; Zhao et al., 2025); however, this method requires vast, costly paired datasets and is difficult to collect and train, and has shown limited practical success. As a result, the dual-tower paradigm has emerged as a more practical alternative. It leverages pre-trained T2V and T2A models, focusing the training effort on an interaction module responsible for fusing audio and video features. The design of this module is critical, with current strategies including Full Attention for direct fusion (Wang et al., 2025b), ControlNet-style (Zhang et al., 2023) conditioning that enables bidirectional influence (i.e., video conditioning audio generation (Liu et al., 2024a), and audio conditioning video (Weng et al., 2025)), and specialized components like the Prior Estimator in Javis-DiT (Liu et al., 2025). Our work adopts the dual-tower paradigm but explore new ways to achieve a more holistic interaction among the text, audio, and video modalities in the T2SV task.

**Visual-Grounded Audio Caption**   Generating accurate captions for audio has long been a challenging task, primarily due to the inherently lower information density of sound compared to other modalities like speech, images, or video (Arandjelović & Zisserman, 2017). Prior approaches (Drossos et al., 2017; Chu et al., 2024) often leverage a single model for direct captioning, but these have been observed to suffer from significant hallucination issues (Sung-Bin et al., 2024; Nishimura et al., 2024; Kuan et al., 2024). Recognizing that audio and visual information are often highly correlated, many human annotation efforts for audio events naturally incorporate visual cues when annotators are uncertain (Kim et al., 2019). This strong interrelation has led to a growing interest in utilizing visual information to assist with audio captioning tasks Arandjelović & Zisserman (2017); Liu et al. (2023c); Kim et al. (2024).

Figure 2: Our three-stage Hierarchical Visual-Grounded Captioning (HVGC) framework generates disentangled modality-pure text captions. First, a Vision-Language Large Model (VLLM) produces a detailed video caption ($T_V$). Subsequently, a Large Language Model (LLM) extracts relevant audio tags from this video caption. Finally, the framework leverages both the visual context in $T_V$ and the extracted audio tags to generate a pure audio caption ($T_A$).

## 3 METHOD

### 3.1 PRELIMINARY

**Generative Models in Denoised Manner** Denoised generative models learn a complex data distribution $p(\mathbf{x})$ by reversing a process that destroys data to a simple Gaussian prior $\mathcal{N}(\mathbf{0}, \mathbf{I})$. Diffusion models (Ho et al., 2020) approach this by training a network $\boldsymbol{\epsilon}_\theta$ to predict the noise $\boldsymbol{\epsilon}$ added to a data sample $\mathbf{x}_0$ at timestep $t$:

$$\mathcal{L}_{\text{DDPM}}(\theta) = \mathbb{E}_{t,\mathbf{x}_0,\boldsymbol{\epsilon}} \left[ ||\boldsymbol{\epsilon} - \boldsymbol{\epsilon}_\theta(\sqrt{\bar{\alpha}_t}\mathbf{x}_0 + \sqrt{1 - \bar{\alpha}_t}\boldsymbol{\epsilon}, t)||^2 \right]. \tag{1}$$

Flow Matching (FM) (Lipman et al., 2022) models learn a velocity field $v_\theta$ that transports a noise sample $\mathbf{x}_0$ to a data sample $\mathbf{x}_1$ by approximating the target field $\mathbf{x}_1 - \mathbf{x}_0$. The training objective is:

$$\mathcal{L}_{\text{FM}}(\theta) = \mathbb{E}_{t,\mathbf{x}_0,\mathbf{x}_1} \left[ ||v_\theta(t\mathbf{x}_0 + (1-t)\mathbf{x}_1, t) - (\mathbf{x}_1 - \mathbf{x}_0)||^2 \right]. \tag{2}$$

Generation in both cases involves starting with a sampled noise and applying the learned network iteratively denoise to obtain a clean data sample. More background is in Appendix B.

**Problem Formulation** For the T2SV task, we adopt the dual-tower paradigm. This approach is highly practical as it leverages the capabilities of pre-trained unimodal models, a video tower $\mathcal{G}_\theta^V$ and an audio tower $\mathcal{G}_\theta^A$. In this setup, the towers independently process their respective text captions, $T_V$ and $T_A$, audio timestep $t_A$ and video timestep $t_V$, noisy audio latent $\mathbf{x}_A(t_A)$ and nosiy video latent $\mathbf{x}_V(t_V)$ while a trainable interaction module, $\mathcal{B}_\theta^{AV}$, facilitates cross-modal communication:

$$(\hat{\mathbf{a}}, \hat{\mathbf{v}}) = \mathcal{G}_{\text{model}}(T_A, T_V, \mathbf{x}_A(t_A), \mathbf{x}_V(t_V), t_A, t_V), \quad \text{where } \mathcal{G}_{\text{model}} = \{\mathcal{G}_\theta^A, \mathcal{G}_\theta^V, \mathcal{B}_\theta^{AV}\}. \tag{3}$$

The final output consists of the predicted audio $\hat{\mathbf{a}}$ and video $\hat{\mathbf{v}}$ noise vector.

**Training Objective** The training objective of T2SV is the sum of the loss from the two towers:

$$\mathcal{L} = \mathcal{L}_{\text{audio}} + \mathcal{L}_{\text{video}}. \tag{4}$$

The audio tower follows a diffusion training setup using a v-prediction diffusion (Salimans & Ho, 2022) loss objective. Given the continuous timestep $t_A \in [0, 1]$, the signal and noise scaling factors are $\alpha(t_A) = \cos(t_A\pi/2)$ and $\sigma(t_A) = \sin(t_A\pi/2)$. We denote $\mathbf{x}_A$ as the audio latent vector from the audio Variational AutoEncoder (VAE) (Kingma & Welling, 2022) encoder. It predicts the target $\alpha(t_A)\boldsymbol{\epsilon}_A - \sigma(t_A)\mathbf{x}_A$ and for the noisy audio latent $\mathbf{x}_A(t_A) = \alpha(t_A)\mathbf{x}_A + \sigma(t_A)\boldsymbol{\epsilon}_A$:

$$\mathcal{L}_{\text{audio}} = \|\hat{\mathbf{a}} - (\alpha(t_A)\boldsymbol{\epsilon}_A - \sigma(t_A)\mathbf{x}_A)\|^2. \tag{5}$$

The video tower follows a flow matching (Lipman et al., 2022) loss objective. The corresponding video timestep $t_V$ is defined as $t_V = 1000 \cdot t_A$. $\mathbf{x}_V$ is the video latent vector. It predicts the target vector field $\boldsymbol{\epsilon}_V - \mathbf{x}_V$ and for the noisy video latent $\mathbf{x}_V(t_V) = (1 - t_V/1000)\mathbf{x}_V + (t_V/1000)\boldsymbol{\epsilon}_V$:

$$\mathcal{L}_{\text{video}} = \|\hat{\mathbf{v}} - (\boldsymbol{\epsilon}_V - \mathbf{x}_V)\|^2. \tag{6}$$

Here, $\boldsymbol{\epsilon}_A$ and $\boldsymbol{\epsilon}_V$ are Gaussian noise vectors sampled from $\mathcal{N}(\mathbf{0}, \mathbf{I})$. The detailed inference process is further shown in Appendix C.3.

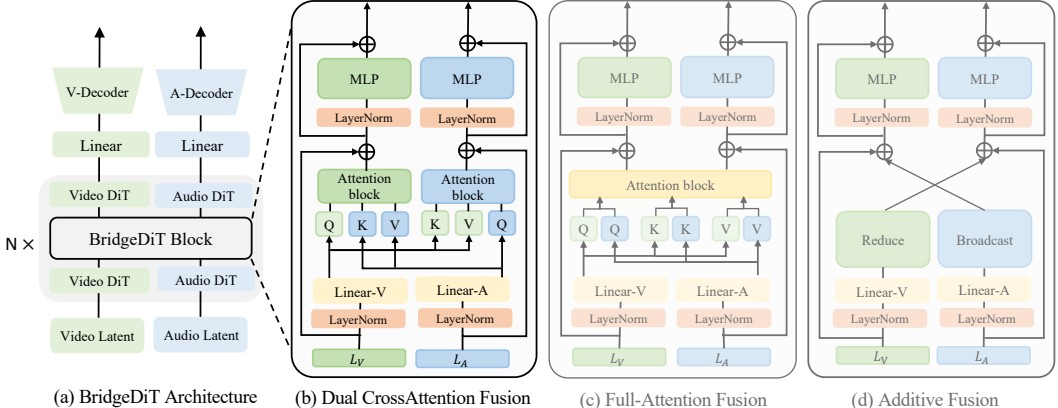

(a) BridgeDiT Architecture    (b) Dual CrossAttention Fusion    (c) Full-Attention Fusion    (d) Additive Fusion

Figure 3: **The BridgeDiT Architecture**. (a): The overall dual-tower architecture. Parallel video and audio DiT streams are connected by our proposed BridgeDiT Block at specific layers. *Right*: Details of fusion strategies within the block, showcasing our proposed Dual Cross-Attention (b) alongside the Full-Attention (c) and Additive Fusion (d) baselines.

### 3.2 HIERARCHICAL VISUAL-GROUNDED CAPTIONING FRAMEWORK

To address the conditioning problem, we introduce the Hierarchical Visual-Grounded Captioning (HVGC) framework. As illustrated in Figure 2, HVGC is a three-stage pipeline designed to generate disentangled, modality-pure video caption ($T_V$) and audio caption ($T_A$) from sounding videos. Since directly generating captions from raw audio, even with advanced Audio Large-Language-Models (LLMs) (Chu et al., 2024), can lead to severe hallucination issues (Sung-Bin et al., 2024; Nishimura et al., 2024; Kuan et al., 2024), for instance, "rhythmic drumming" might be misinterpreted as "high heels clicking on a pavement". This is due to the ambiguity of information conveyed by audio. HVGC tackles this by grounding audio caption generation in a rich visual context.

Initially (Stage 1), a powerful Vision-Language Large Language Model (VLLM), such as Qwen2.5-VL-72B (Bai et al., 2025), produces a comprehensive visual description ($T_V$) of the video clip. We employ an in-context learning approach with a meticulously designed prompt. This prompt guides the VLLM to detail the video's environment, subject actions, cinematography, and overall style. Subsequently (Stage 2), an auxiliary Large Language Model (LLM) abstracts key auditory event tags (e.g., 'hammer striking metal', 'hiss of sparks') directly from $T_V$. This process, inspired by Chain-of-Thought (CoT) prompting (Wei et al., 2022; Teng et al., 2025), acts as an intermediate filter, distilling the visual context into relevant sound-producing elements, thereby preventing the final audio caption from including non-existent sounds. Finally (Stage 3), leveraging both the detailed video caption($T_V$) and the abstracted auditory tags, we use an LLM (Qwen2.5-72B (Yang et al., 2024)) to generate the final audio caption ($T_A$). This crucial step ensures $T_A$ is not only contextually consistent with the video narrative but also articulated exclusively using non-visual, auditory language. This hierarchical, visually-grounded approach delivers pure unimodal captions, effectively eliminating cross-modal interference for optimal performance of our dual-tower T2SV model. Detailed prompts for HVGC are provided in the Appendix C.6.

### 3.3 THE BRIDGEDIT ARCHITECTURE

To address the interaction problem, we introduce BridgeDiT, a novel dual-tower diffusion transformer architecture for sounding video generation. As depicted in Figure 3 (a), BridgeDiT consists of two parallel pre-trained DiT backbones for video and audio that remain largely frozen. To thoroughly investigate the optimal strategy for effective cross-modal fusion, we propose Dual Cross-Attention (DCA) fusion mechanism within each BridgeDiT Block. We compare DCA against several alternative fusion mechanisms with detailed experiments presented in Section 4.3.2. Furthermore, an ablation study on the optimal placement of BridgeDiT Blocks across different layers is discussed in Appendix D.

**Dual CrossAttention Fusion** As detailed in Figure 3 (b), similar to dual cross-attention architectures used in other multimodal tasks such as video generation (Liu et al., 2023a) and multimodal understanding (Nguyen et al., 2022; Wang et al., 2022; Tsai et al., 2019; Lu et al., 2019; Yun et al.,

2021), our fusion mechanism takes the video latent $L_V$ and the audio latent $L_A$. Features are then updated through two parallel, symmetric streams within the block. In the Audio-to-Video (A-to-V) stream, video features are refined based on audio context. For this operation, the video latent $L_V$ first passes through a Layer Normalization (LN) layer and is then projected by 'Linear-V' to form the query (Q). Concurrently, the audio latent $L_A$ also undergoes Layer Normalization and is then projected by 'Linear-A' to provide the key (K) and value (V). Here, $LN(\cdot)$ denotes Layer Normalization:

$$Q_V = \text{Linear}_{Q_V}(\text{LN}(L_V)), \quad K_A = \text{Linear}_{K_A}(\text{LN}(L_A)), \quad V_A = \text{Linear}_{V_A}(\text{LN}(L_A)) \quad (7)$$

The resulting projections $(Q_V, K_A, V_A)$ are fed into a cross-attention layer. The output of this attention operation is then integrated back into the video latent via a residual connection to produce the updated video latent, $L'_V$:

$$L'_V = \text{Attention}(Q_V, K_A, V_A) + L_V \quad (8)$$

This is subsequently passed through a Layer Normalization and an MLP block with another residual connection, completing the video feature update. Concurrently, the Video-to-Audio (V-to-A) stream operates in a perfectly symmetric manner. In this case, the audio latent $L_A$ serves as the query, while the video latent $L_V$ provides the key and value. The update process is analogous, yielding the updated audio latent, $L'_A$. Consistent with the DiT (Peebles & Xie, 2022) paradigm, the BridgeDiT Block is also conditioned on the timestep condition ($t_V$ and $t_A$) via the adaptive layer normalization (AdaLN) (Perez et al., 2018) mechanism.

**Alternative Fusion Mechanism** To thoroughly validate the effectiveness of our Dual Cross-Attention (DCA) fusion mechanism, we compare it against several alternative fusion strategies adopted from existing works. These baselines, also visualized in Figure 3 alongside our DCA, are implemented under the same settings as the BridgeDiT Block.

- **Full Attention Fusion:** As shown in Figure 3 (c), this method performs a joint self-attention operation across both modalities. First, the video latent $L_V$ and audio latent $L_A$ are independently projected into query, key, and value representations after normalization. These modality-specific projections are then concatenated along the sequence dimension to form unified tensors:

$$Q_{\text{cat}} = \text{Concat}(Q_V, Q_A), \quad K_{\text{cat}} = \text{Concat}(K_V, K_A), \quad V_{\text{cat}} = \text{Concat}(V_V, V_A). \quad (9)$$

A single self-attention operation is applied to these unified tensors, allowing for all-to-all interaction. The output is then residually connected with the original concatenated latents:

$$L'_{\text{cat}} = \text{Attention}(Q_{\text{cat}}, K_{\text{cat}}, V_{\text{cat}}) + \text{Concat}(L_V, L_A). \quad (10)$$

Finally, this fused representation is split back into separate video and audio latents, $L'_V$ and $L'_A$. JoinDiT (Wang et al., 2025b) use this fusion for image-conditioned sound video generation.

- **Additive Fusion:** As shown in Figure 3 (d), this method uses a highly efficient and lightweight alternative that projects and combines video and audio feature with element-wise addition. Due to its small parameter count, this method was adapted by SSVG (Ishii et al., 2024).

- **Unidirectional Cross-Attention:** This approach treats one modality as the condition for the other in a ControlNet-style (Zhang et al., 2023). In our baseline implementations, the tower that provides the condition processes its own features using a standard self-attention block, while the other tower uses the same cross-attention block as our DCA fusion mechanism. We implement both V2A (Liu et al., 2024a) and A2V (Weng et al., 2025) variants for comparison. (The architecture figures are omitted for brevity, as the overall structure is similar to DCA)

## 4 EXPERIMENTS

### 4.1 EXPERIMENT SETUP

**Implementation Details** For the video backbone, we utilize the WAN 2.1 (1.3B) model (Wan et al., 2025), retaining its original configuration with a UMT5-XXL (Raffel et al., 2020) text encoder to generate 81 frames at 15fps and a 480p resolution. For the audio backbone, we employ Stable

Table 1: Automatic evaluation on the AVSync15 dataset. **Best** and second-best are highlighted.

| Process | Method | Quality | | | | Text Alignment | | Synchronization | |
|---------|--------|---------|---------|--------|--------|----------|--------|--------|-----------|
| | | FVD↓ | KVD↓ | FAD↓ | KL↓ | CLIPSIM↑ | CLAP↑ | VA-IB↑ | AV-Align↑ |
| T → V ‖ T → A | Wan + SDA | 828.33 | 22.56 | 11.90 | 3.17 | 28.12 | 30.78 | 26.22 | 0.205 |
| T → V + V → A | Wan + MMAudio | 828.33 | 22.56 | 7.98 | 1.40 | 28.12 | 34.64 | 33.50 | 0.243 |
| | Wan + SeeingHearing | 828.33 | 22.56 | 14.21 | 2.89 | 28.12 | 25.52 | **35.87** | 0.208 |
| T → A + A → V | SDA + TPos | 1975.22 | 92.95 | 11.90 | 3.17 | 18.64 | 30.78 | 25.45 | 0.176 |
| | SDA + TempoToken | 1516.53 | 42.30 | 11.90 | 3.17 | 20.56 | 30.78 | 17.63 | 0.215 |
| T → I + I → AV | JointDiT | 992.71 | 25.20 | 6.51 | 1.77 | **29.94** | 30.34 | 34.17 | 0.156 |
| | JavisDiT | 878.70 | 23.23 | 13.48 | 3.50 | 28.05 | 22.99 | 20.75 | 0.158 |
| | SSVG | 1028.78 | 31.44 | 14.35 | 4.35 | 25.78 | 23.67 | 22.45 | 0.126 |
| T → VA | MTV | 982.09 | 24.60 | 16.46 | 3.53 | 27.66 | 21.22 | 15.84 | 0.149 |
| | CoDi | 1387.14 | 39.24 | 16.56 | 5.24 | 24.96 | 17.94 | 10.79 | 0.081 |
| | **BridgeDiT (ours)** | **765.74** | **21.33** | **5.34** | **1.30** | 28.52 | **35.95** | 34.59 | **0.275** |

Audio Open 1.0 (Evans et al., 2025) with a T5-base text encoder (Raffel et al., 2020), generating audio at a 44.1kHz sample rate. The total generation length is standardized to 5.4 seconds. Our BridgeDiT architecture consists of 4 BridgeDiT Blocks, which are uniformly inserted between the corresponding layers of the video and audio towers. We trained our models separately for each dataset. To ensure a fair comparison, we categorized the baselines into two types. We utilized the publicly available pre-trained models for inference without additional training. These models were already trained on extensive datasets, such as VGGSound and Landscape, making further training on our specific benchmarks redundant. For the baselines we constructed, all models were trained using our HVGC captions. This ensures consistency in captioning across these baselines, allowing for a fair evaluation of the performance gains attributable to our proposed architecture. More details are in the Appendix C.

**Dataset** We evaluate our model on the T2SV task using three datasets: AVSync15 (Zhang et al., 2024), VGGSound-SS (Chen et al., 2021), and Landscape (Lee et al., 2022). (1) AVSync15 (Zhang et al., 2024) is a subset of VGGSound Chen et al. (2020) and contains synchronized audio-video pairs across 15 categories. The dataset is split into 1350 videos for training and 150 for testing. (2) VGGSound-SS (Chen et al., 2021) is a sound source localization dataset, also derived from VGGSound (Chen et al., 2020), where the sounding object is always visually present. It includes 5,158 videos from 220 different classes. We randomly sample 150 videos to form our test set. (3) Landscape (Lee et al., 2022): This dataset comprises 928 videos, depicting 9 different scenic categories. Since the official versions of these datasets lack standard captions, we generated them using HVGC. For preprocessing, we first ensure audio-visual correspondence by retaining only pairs with an ImageBind (Girdhar et al., 2023) score above 0.3. Subsequently, all videos are standardized to a 5.4-second duration via random cropping or padding.

**Baseline** We compare our method against a comprehensive set of baselines, which we categorize into five distinct T2SV generation strategies. (1) T → V ‖ T → A: This baseline generates video and audio independently. To ensure a fair comparison, we implement this by disabling the interaction modules and only training two separate towers, as Wan+SDA. (2) T → V → A: This pipeline first generates a video from text and subsequently generates

Table 2: Performance on VGGSound-SS and Landscape. AV denotes AV-Align metric here. **Best** and second-best are highlighted.

| Method | VGGSound-SS | | | LandScape | | |
|--------|-------------|--------|-----------|-----------|--------|-----------|
| | FVD↓ | FAD↓ | AV-Align↑ | FVD↓ | FAD↓ | AV-Align↑ |
| *T → V ‖ T → A* | | | | | | |
| Wan + SDA | 737.96 | 8.05 | 0.238 | 700.18 | 5.80 | 0.177 |
| *T → V + T → A* | | | | | | |
| Wan + MMAudio | 737.96 | **5.39** | 0.333 | 700.18 | 5.31 | 0.218 |
| Wan + SeeingHearing | 737.96 | 8.02 | 0.321 | 700.18 | 7.59 | 0.166 |
| *T → A + A → V* | | | | | | |
| SDA + TPos | 1732.47 | 8.05 | 0.163 | 1837.42 | 5.80 | 0.143 |
| SDA + TempoToken | 1942.94 | 8.05 | 0.242 | 2089.05 | 5.80 | 0.206 |
| *T → I + I → VA* | | | | | | |
| JointDiT | 866.59 | 7.19 | 0.125 | 937.09 | 6.35 | 0.075 |
| *T → VA* | | | | | | |
| JavisDiT | 637.50 | 7.78 | 0.179 | 668.87 | 9.22 | 0.185 |
| SSVG | 1032.87 | 8.86 | 0.148 | 1186.39 | 8.97 | 0.143 |
| CoDi | 1203.68 | 8.38 | 0.113 | 1220.31 | 13.75 | 0.082 |
| **BridgeDiT (ours)** | **615.28** | 6.01 | **0.3617** | **628.07** | **4.78** | **0.258** |

Table 3: Ablation study on disentangled text condition. We compare shared caption strategies (using the video caption or an Omini model caption) against disentangled caption strategies (using an Audio-LLM, Omini Model or our method) in both full-training and zero-shot settings. We also do an ablation study on different stages of HVGV.

| Caption Source | If-Training | FVD ↓ | FAD ↓ | CLIPSIM ↑ | CLAP ↑ | VA-IB ↑ | AV-Align ↑ |
|---|---|---|---|---|---|---|---|
| *Shared text condition ($T_V = T_A$)* | | | | | | | |
| Video Caption ($T_V$) | No | 908.81 | 16.39 | 28.34 | 9.65 | 18.92 | 0.172 |
| Omini Caption | No | 1362.83 | 13.75 | 25.81 | 22.45 | 23.36 | 0.135 |
| Video Caption ($T_V$) | Yes | 828.33 | 7.75 | 28.31 | 25.10 | 29.22 | 0.245 |
| Omini Caption | Yes | 1232.46 | 7.56 | 25.76 | 31.16 | 33.76 | 0.229 |
| *Disentangled text condition ($T_V \neq T_A$)* | | | | | | | |
| Omini Caption ($T_A$ and $T_V$) | No | 1423.48 | 16.89 | 25.37 | 23.86 | 23.89 | 0.157 |
| HVGC (No Stage 2) | No | 908.81 | 15.48 | 28.34 | 25.48 | 23.32 | 0.198 |
| $T_A$ from AudioLLM | No | 908.81 | 16.36 | 28.34 | 20.62 | 15.86 | 0.135 |
| HVGC | No | 908.81 | 14.90 | 28.34 | 28.37 | 25.36 | 0.211 |
| $T_A$ from AudioLLM | Yes | 776.25 | 12.12 | 26.35 | 27.76 | 19.44 | 0.242 |
| **HVGC (ours)** | **Yes** | **765.74** | **5.34** | **28.52** | **35.95** | **34.59** | **0.275** |

the audio conditioned on the video. We employ Wan-1.3B for the T2V step, followed by the V2A models MMAudio (Cheng et al., 2025a) and SeeingHearing (Xing et al., 2024). (3) T → A → V: This method first generates audio from text using Stable Diffusion Audio Open (SDA) (Evans et al., 2025), then generates a video conditioned on this audio using the T-Pos (Jeong et al., 2023) and TempoToken (Cao et al., 2023). (4) T → I → AV: This approach uses an intermediate image generated by Qwen-Image (Wu et al., 2025). Then, the JointDiT (Wang et al., 2025b) model is used to jointly generate the video and audio from this image. (5) T → AV: This strategy includes existing joint-training models such as JavisDiT (Liu et al., 2025), SSVG (Ishii et al., 2024), CoDi (Tang et al., 2023), and MTV (Weng et al., 2025). More details about these baselines are in the Appendix C.2.

**Evaluation Metric** We evaluate T2SV from three different perspectives: generation quality, text alignment, and audio-video synchronization. (1) Generation Quality. For video quality, we employ the Fréchet Video Distance (FVD) (Unterthiner et al., 2018) and Kernel Video Distance (KVD) (Unterthiner et al., 2018). For audio quality, we use the Fréchet Audio Distance (Kilgour et al., 2018) (FAD) and the Kullback-Leibler (Wang et al., 2024b) (KL) divergence score. (2) Text Alignment. We evaluate video and audio text alignment separately. We use CLIPSIM (Radford et al., 2021) to evaluate video-text alignment and CLAP (Elizalde et al., 2023) score to measure audio-text alignment. (3) Audio-Video Synchronization. We evaluate both semantic sync using the ImageBind score (IB-VA) (Girdhar et al., 2023) and temporal sync using the AV-Align score (Yariv et al., 2024).

## 4.2 Main Results: Comparison with Baselines

We compare our propose approach with baselines on three datasets and present results in Table 1 and Table 2, demonstrating that our approach surpasses all baselines on most metrics, including video quality (FVD, KVD), audio quality (FAD, KL), audio-text alignment (CLAP), and temporal synchronization (AV-Align). First, our model significantly outperforms the Wan+SDA baseline, which is equivalent to our architecture but with the interaction modules removed. This validates the effectiveness of our BridgeDiT Block, proving that enabling cross-modal interaction is crucial for enhancing the generative quality of both modalities and achieving strong semantic and temporal synchronization. Second, BridgeDiT consistently outperforms pipelined baselines , which suggests that our joint generation approach effectively mitigates the error accumulation inherent in pipeline strategy. We observe two minor exceptions. Our CLIPSIM score (28.52) is slightly lower than that of JointDiT (Wang et al., 2025b) (29.94), a gap we attribute to better alignment with T2I backbone Qwen-Image (Wu et al., 2025). Our IB-VA score (34.59) is also surpassed by SeeingHearing (35.87), which is expected as SeeingHearing model uses ImageBind Score (Girdhar et al., 2023) as classifier guidance. Finally, as shown in Table 2, BridgeDiT also achieves state-of-the-art results on most metrics on the VGGSound-SS (Chen et al., 2021) and Landscape (Lee et al., 2022) datasets, confirming its strong generalization capability.

## 4.3 ABLATION STUDIES

### 4.3.1 EFFECT OF DISENTANGLED TEXTUAL CONDITIONING

We compare HVGC against several caption strategies as shown in Table 3: shared captions (using either video caption for both towers or a single caption from Omini Model like Qwen2.5-Omini (Xu et al., 2025)) and disentangled captions (generating $T_A$ with an Audio LLM like Qwen2-Audio (Chu et al., 2024)). From the results, we derive three key insights: (1) Our HVGC framework consistently yields the best performance across both zero-shot (without training the interaction module) and full-training settings, demonstrating its robust superiority. (2) Within the shared text condition setting, the Omini-model caption improves audio-related metrics (FAD, CLAP) but harms video quality and synchronization. This highlights the inherent limitation of a single shared caption to adequately serve both modalities. (3) The alternative disentangled baseline, which uses an Audio LLM for the audio caption ($T_A$), performs poorly. This is due to significant hallucination issues, where the model invents sounds inconsistent with the visual scene, thereby degrading overall performance (see Appendix C.8 for examples). We also ablate the necessity of Stage 2 by comparing against a variant that removes Stage 2. This confirms that our multi-stage design is a necessary process to generate accurate, hallucination-free audio captions.

### 4.3.2 ANALYSIS OF FUSION MECHANISMS

To investigate the optimal architecture for cross-modal interaction, we conduct an ablation study on different fusion mechanisms, as illustrated in Figure 4. We compared our proposed Dual CrossAttention (DCA) fusion mechanism against several existing fusion mechanisms: Full-Attention Fusion (FullAttn) (Wang et al., 2025b), Additive fusion (Add-Fusion) (Ishii et al., 2024), and two unidirectional cross-attention variants (V2A-CrossAttn (Liu et al., 2024a) and A2V-CrossAttn (Weng et al., 2025)). We measure AV-Align (Yariv et al., 2024) (top) and VA-IB Score (Girdhar et al., 2023) (bottom) over the course of training. The results clearly demonstrate that our DCA fusion mechanism consistently outperforms all other fusion mechanisms. It achieves the highest scores in both AV-Align and VA-IB Score throughout the training process, indicating superior temporal and semantic synchronization. The second-best method is FullAttn, which allows for expressive and all-to-all feature interaction. The unidirectional cross-attention methods (V2A-CrossAttn, A2V-CrossAttn) and additive fusion (Add-Fusion) show comparatively weaker performance. All the experiments underscore our key insight: an effective and efficient bidirectional information exchange is critical for achieving state-of-the-art audio-video synchronization.

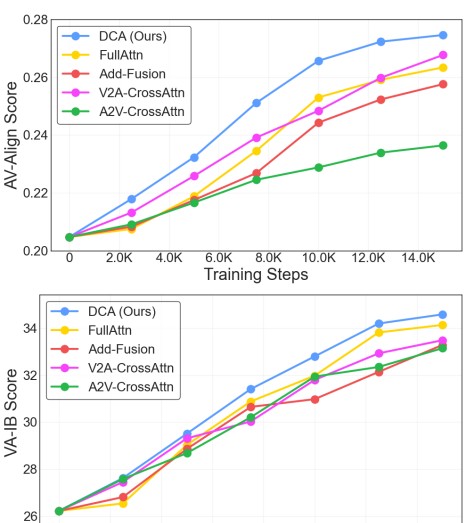

Figure 4: Comparing different fusion mechanisms. Our DCA fusion mechanism outperforms all other baselines in both AV-Align and VA-IB Score.

## 4.4 CASE STUDIES

Figure 1 presents case studies that highlight the capabilities of our BridgeDiT model. Powered by the combination of our HVGC framework and the DCA fusion mechanism, our model generates high-quality sound videos that are semantically synchronized, temporally synchronized, and highly aligned with the text conditions. The first case (the blacksmith) showcases precise temporal synchronization, as the visual impact of the hammer striking the iron aligns perfectly with the sharp "clang" event in the audio spectrogram. The second case (the saxophone player) demonstrates strong text alignment; the generated video accurately depicts key entities from the visual prompt, including the "saxophone" and "metal drum", while the audio faithfully synthesizes the complex soundscape described in the audio prompt. These examples are representative of our model's performance, and we provide a more comprehensive collection of generated videos on our anonymous demo page [1].

---

[1] https://bridgedit-t2sv.github.io

## 4.5 USER STUDY

To further validate our model with human preference, we conducted a user study on the AVSync15 test set with 150 samples. 20 evaluators rate the generated sounding videos on a 0-5 scale with 0.5-point increments across five criteria: Video Quality (VQ), Audio Quality (AQ), Text Alignment (TA), Synchronization (Sync), and Overall quality. As shown in Table 4, our BridgeDiT model is rated highest across all five dimensions, significantly outper-

Table 4: User study on AVSync15 test set with 20 raters

| Method | VQ ↑ | AQ↑ | TA↑ | Sync↑ | Overall↑ |
|---|---|---|---|---|---|
| Wan+SDA | 3.16 | 2.93 | 2.75 | 2.47 | 2.79 |
| Wan+Seeing | 3.16 | 2.83 | 2.91 | 2.54 | 2.85 |
| Wan+MMAudio | 3.16 | 3.18 | 3.07 | 2.77 | 3.06 |
| SDA+TmToken | 1.97 | 2.36 | 2.00 | 1.83 | 2.04 |
| JointDiT | 2.74 | 2.81 | 2.79 | 2.45 | 2.69 |
| JavisDiT | 2.47 | 2.39 | 2.36 | 2.14 | 2.36 |
| **BridgeDiT (ours)** | **3.40** | **3.33** | **3.33** | **3.09** | **3.34** |

forming all baselines, while the pipelined Wan + MMAudio method ranked as the second-best performer. Notably, while some baselines may achieve superior results on certain metrics (as in Table 1), our model remains the leading preference in human evaluations. This suggests that automatic metrics may not fully align with human preference.

## 5 CONCLUSION

In this work, we address two fundamental challenges in T2SV generation: the condition problem caused by shared text caption and the interaction problem in dual-tower architectures. We introduce the Hierarchical Visual-Grounded Captioning (HVGC) framework to generate disentangled, modality-pure captions and the BridgeDiT architecture with its Dual CrossAttention mechanism for symmetric and efficient fusion. Through comprehensive experiments, our method achieves state-of-the-art performance, a result supported by both automatic metrics and human evaluations. Finally, our detailed ablation studies validate the effectiveness of each proposed component and offer valuable insights for the design of future T2SV models. Limitations and future work direction are discussed in Appendix F.

## ETHICS STATEMENT

We acknowledge that Text-to-Sounding-Video generation technology, like other generative technologies, carries potential risks of misuse. The ability to create realistic and synchronized audio-visual content from text could be exploited to generate convincing disinformation and fraudulent materials. The primary motivation for our research, however, is positive. We believe this technology holds significant potential for beneficial applications. We are committed to the responsible advancement of this field and encourage continued research into synthetic content detection and the establishment of clear ethical guidelines for deployment.

## REPRODUCIBILITY STATEMENT

To ensure the reproducibility of our work, detailed experimental information can be found in Appendix C (including Compute Resources C.1, Baselines C.2, Inference Process Details C.3, Hyperparameters C.4, HVGC Prompts C.6, Human Annotation Command C.7, and Caption Examples C.8). Furthermore, the complete source code, trained model checkpoints, and datasets necessary to replicate our results will be made publicly available at `https://bridgedit-t2sv.github.io/`. We are committed to transparency and facilitating future research in this area.

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

## A  THE USE OF LARGE LANGUAGE MODELS

In this work, Large Language Models (LLMs) are used solely for enhancing writing clarity and English expression. All core contributions, including model design, mathematical formulations, and experimental analyses, are developed independently by the authors. The authors take full responsibility for the final content, ensuring no plagiarism or fabrication occurred.

## B  DIFFUSION AND FLOW MATCHING GENERATION MODELS

Generative models are designed to learn a complex data distribution $p(\mathbf{x})$ from a simple prior, typically a standard Gaussian distribution $\mathcal{N}(\mathbf{0}, \mathbf{I})$. Many state-of-the-art approaches are based on learning to reverse a predefined process that maps data to noise. A prominent family of such models is diffusion models. In their foundational formulation (DDPM) (Ho et al., 2020), they utilize a fixed forward process that progressively adds Gaussian noise to a data sample $\mathbf{x}_0$ over discrete timesteps. The resulting noisy sample at any time $t$, denoted $\mathbf{x}_t$, can be expressed as $\mathbf{x}_t = \sqrt{\bar{\alpha}_t}\mathbf{x}_0 + \sqrt{1 - \bar{\alpha}_t}\boldsymbol{\epsilon}$, where $\bar{\alpha}_t$ is a predefined noise schedule and $\boldsymbol{\epsilon} \sim \mathcal{N}(\mathbf{0}, \mathbf{I})$. A neural network, $\boldsymbol{\epsilon}_\theta(\mathbf{x}_t, t)$, is then trained to predict the noise component $\boldsymbol{\epsilon}$ from the corrupted sample:

$$L_{\text{DDPM}}(\theta) = \mathbb{E}_{t, \mathbf{x}_0, \boldsymbol{\epsilon}}\left[||\boldsymbol{\epsilon} - \boldsymbol{\epsilon}_\theta(\mathbf{x}_t, t)||^2\right]$$

More recent frameworks, such as EDM (Karras et al., 2022), generalize this process by formulating it as solving a continuous-time stochastic differential equation (SDE). EDM provides a principled design methodology, emphasizing crucial choices like network preconditioning. The denoiser network, $D_\theta(\mathbf{x}_t, \sigma_t)$, is scaled to have consistent input and output magnitudes across all noise levels $\sigma_t$. This network is often trained to predict the clean data $\mathbf{x}_0$ directly, using a weighted loss function that prioritizes different noise levels:

$$L_{\text{EDM}}(\theta) = \mathbb{E}_{t, \mathbf{x}_0, \boldsymbol{\epsilon}}\left[\lambda(\sigma_t)||D_\theta(\mathbf{x}_t, \sigma_t) - \mathbf{x}_0||^2\right]$$

For the generation part, both approaches start with a sample from the prior, $\mathbf{x}_T \sim \mathcal{N}(\mathbf{0}, \mathbf{I})$, and iteratively apply the learned denoising function to recover a clean sample $\mathbf{x}_0$. As an alternative to the noise-prediction framework, Flow Matching (FM) (Lipman et al., 2022) models learn to generate data in a single continuous-time transformation. These models learn a vector field $\mathbf{v}_t$ that transports samples from a prior distribution $p_0$ (noise) to the target data distribution $p_1$ (data) by following an ordinary differential equation (ODE): $\frac{d\mathbf{x}_t}{dt} = \mathbf{v}_t(\mathbf{x}_t)$. To make training tractable, FM trains a network $v_\theta$ to approximate a simple, predefined vector field. For a linear path between a noise sample $\mathbf{x}_0 \sim p_0$ and a data sample $\mathbf{x}_1 \sim p_1$, the target vector field is simply their difference, $\mathbf{x}_1 - \mathbf{x}_0$. The corresponding FM loss is:

$$L_{\text{FM}}(\theta) = \mathbb{E}_{t, \mathbf{x}_0, \mathbf{x}_1}\left[||v_\theta(t, (1-t)\mathbf{x}_0 + t\mathbf{x}_1) - (\mathbf{x}_1 - \mathbf{x}_0)||^2\right]$$

To generate a sample, one simply solves the learned ODE $\frac{d\mathbf{x}_t}{dt} = v_\theta(t, \mathbf{x}_t)$ from $t = 0$ to $t = 1$, starting with an initial sample $\mathbf{x}_0 \sim p_0$.

**Classifier-Free Guidance**  Conditional generation in these models is commonly achieved using Classifier-Free Guidance (CFG) (Ho & Salimans, 2022). This technique steers the generation process towards a desired condition $c$ (e.g., a text prompt) without needing an external classifier. The model, here denoted with the noise predictor $\boldsymbol{\epsilon}_\theta(\mathbf{x}_t, t, c)$, is jointly trained on conditional inputs $c$ and a null token $\emptyset$. During sampling, the guided prediction $\hat{\boldsymbol{\epsilon}}_\theta$ is an extrapolation from the unconditional prediction towards the conditional one:

$$\hat{\boldsymbol{\epsilon}}_\theta = \boldsymbol{\epsilon}_\theta(\mathbf{x}_t, t, \emptyset) + w(\boldsymbol{\epsilon}_\theta(\mathbf{x}_t, t, c) - \boldsymbol{\epsilon}_\theta(\mathbf{x}_t, t, \emptyset))$$

The guidance scale $w > 1$ is a hyperparameter that adjusts the strength of the condition. A larger $w$ typically improves fidelity to the condition at the cost of reduced sample diversity. This technique is applied analogously to other model predictions like $D_\theta$ or $v_\theta$.

## C  EXPERIMENTS SETUP

### C.1  COMPUTE RESOURCES

All experiments in this work were conducted on 4 nodes equipped with NVIDIA H100 80GB GPUs. Each node further utilized 64 Intel(R) Xeon(R) Platinum 8481C CPUs @ 2.70GHz, with 2TB of

RAM and 4TB of SSD storage. For generating the high-quality examples presented in our anonymous demo page, we utilized 2 NVIDIA B200 180GB GPU nodes. On these nodes, we replaced our standard video generation backbone with the more advanced Wan14B model to achieve superior visual fidelity, while maintaining identical CPU, RAM, and storage specifications per node.

## C.2 BASELINES

Here we detail the baseline models used in our work.

- **Wan** (Wan et al., 2025) is a large-scale video generative model (available in 1.3B and 14B versions) renowned for producing high-resolution and temporally coherent videos, representing a leading open-source T2V model.
- **Stable-Audio-Open** (Evans et al., 2025) is a diffusion-based text-to-audio generation model trained on a large dataset to create diverse and realistic audio content.
- **MMAudio** (Cheng et al., 2025a) is a video-to-audio synthesis model designed to generate synchronized sound for silent video clips.
- **Seeing-and-Hearing** (Xing et al., 2024) introduces "diffusion latent aligners" that leverage the ImageBind embedding space to create a shared latent space for visual and auditory data, enabling semantic alignment guidance.na
- **TPos** (Jeong et al., 2023) focuses on audio-reactive video generation, creating dynamic and visually engaging videos that respond to the rhythm and emotional tone of an input audio track.
- **TempoToken** (Cao et al., 2023) proposes "TempoTokens," learnable embeddings that guide audio-to-video generation, ensuring both temporal alignment between audio and visual output.
- **JointDiT** (Wang et al., 2025b) is a dual-tower joint generative model for image-conditioned sound video generation. It employs a Full Attention fusion mechanism, though its performance can be limited by its T2V backbone (e.g., Stable Video Diffusion).
- **JavisDiT** (Liu et al., 2025) is a Joint Audio-Video Diffusion Transformer (JAVG) built on the DiT architecture. It achieves high-quality, synchronized audio-video generation from open-ended prompts by introducing a Hierarchical Spatio-Temporal Synchronized Prior (HiST-Sypo) Estimator for fine-grained alignment.
- **SSVG** (Ishii et al., 2024) presents a simple yet strong baseline for sounding video generation. It integrates base audio and video diffusion models with novel mechanisms like timestep adjustment and Cross-Modal Conditioning as Positional Encoding (CMC-PE), which is an additive-fusion mechanism to enhance audio-video alignment.
- **MTV** (Weng et al., 2025) is a versatile framework for audio-sync video generation that explicitly separates audio into speech, effects, and music tracks. This enables disentangled control over lip motion, event timing, and visual mood, leading to fine-grained and semantically aligned video generation. It also introduces the DEMIX dataset.
- **CoDi** (Tang et al., 2023) (Composable Diffusion) is a versatile any-to-any generation model that composes diffusion models trained on different modalities to handle various input and output modalities, including text, images, video, and audio.

## C.3 INFERENCE OF OUR BRIDGEDIT MODEL

For both video and audio generation, we apply Classifier-Free Guidance independently, leveraging separate guidance scales for each modality to fine-tune their respective generation quality and adherence to the text prompts. The guided noise prediction for each modality is given by:

$$\hat{\epsilon}_v(\mathbf{x}_v, T_V) = \epsilon_v(\mathbf{x}_v, \emptyset) + w_v \cdot (\epsilon_v(\mathbf{x}_v, T_V) - \epsilon_v(\mathbf{x}_v, \emptyset)) \tag{11}$$

$$\hat{\epsilon}_a(\mathbf{x}_a, T_A) = \epsilon_a(\mathbf{x}_a, \emptyset) + w_a \cdot (\epsilon_a(\mathbf{x}_a, T_A) - \epsilon_a(\mathbf{x}_a, \emptyset)) \tag{12}$$

Here, $\mathbf{x}_v$ and $\mathbf{x}_a$ represent the noisy video and audio latents at a given timestep, respectively. $\epsilon_v(\mathbf{x}_v, T_V)$ and $\epsilon_a(\mathbf{x}_a, T_A)$ are the predictions from the BridgeDiT model conditioned on their respective text prompts, while $\epsilon_v(\mathbf{x}_v, \emptyset)$ and $\epsilon_a(\mathbf{x}_a, \emptyset)$ are predictions from unconditioned (null) prompts. $w_v$ and $w_a$ are the video and audio guidance scales, allowing for independent control over the trade-off between sample quality and text alignment for each modality.

## C.4 HYPERPARAMETERS

Table 5: Key hyperparameters for our BridgeDiT model.

| Parameter | Value |
|---|---|
| *Training Configuration* | |
| Optimizer | AdamW |
| Learning Rate | 5e-5 |
| Weight Decay | 1e-3 |
| Adam Betas | (0.9, 0.95) |
| LR Scheduler | Cosine decay with linear warmup |
| LR Warmup Steps | 1,000 |
| Total Training Steps | 15,000 |
| Minimum Learning Rate | 1e-6 |
| Unconditional Probability (CFG) | 0.1 |
| Training Precision | bfloat16 |
| *Architecture Configuration* | |
| Number of BridgeDiT Blocks | 4 |
| BridgeDiT Block Channels (Q, K, V) | 1536 |
| BridgeDiT Block Heads | 12 |
| BridgeDiT Timestep Embedding Dim | 1536 |
| Video Tower Bridge Points (Layers) | [3, 7, 11, 15] |
| Audio Tower Bridge Points (Layers) | [2, 5, 8, 11] |
| Trainable Layers (Video) | Last 5 blocks |
| Trainable Layers (Audio) | Last 5 blocks |
| *Sampling Configuration* | |
| Video Resolution | $834 \times 480$ |
| Video Number of Frames | 81 |
| Video Frame Rate (fps) | 15 |
| Audio Sample Rate | 44100 Hz |
| Audio Duration | 5.4 seconds |
| Number of Inference Steps | 50 |
| Video Guidance Scale (CFG) | 6.0 |

## C.5 DETAILS ABOUT SPATIAL INFORMATION IN THE AUDIO

Sounding video with stereo audio is a highly promising direction. However, current sounding video generation models, including Veo 3 and Sora 2, primarily focus on mono audio. Furthermore, existing academic datasets are currently mono-only and do not support stereo audio. Therefore, in this paper, we exclusively consider mono audio generation. When utilizing Stable Audio Open, we handle this by duplicating the single audio track for stereo output.

## C.6 PROMPTS OF OUR HIERARCHICAL VISUAL-GROUNDED CAPTIONING (HVGC) FRAMEWORK

***Instruction for Stage1: Detailed Visual Scene Description***

You are an expert video analyst and prompt engineer. Your goal is to watch a video and create a highly effective, descriptive prompt that can be used by a text-to-video generation model to recreate the visual essence and physical dynamics of the scene.

Your generated prompt should be a dense, continuous paragraph rich with visual details. Follow this thinking process:

1. **Scene and Atmosphere**:
* Describe the core environment and the overall mood. Translate any auditory feeling (e.g., tension from music) into visual terms (e.g., `high-contrast lighting, deep shadows`).
2. **Subjects and Details**:
* Identify the main subjects and objects. Describe them with specific visual adjectives (`a weathered blacksmith with soot-stained hands`, `a glowing orange piece of iron`).
3. **Key Actions, Cinematography, and Physical Dynamics**:
* Describe the sequence of most important actions.
* Describe the cinematography (shot type, angle, movement).
* **[NEW CORE RULE] Describe the "Visual Counterpart" of Sound**: Instead of describing sound itself, describe the physical actions that *create* sound. Focus on impact, interaction, and motion that implies sound.
* **For Speech/Vocalization**: Detail the mouth movements, facial expressions, and throat or chest movements (`a lion opens its massive jaws wide, a deep roar building in its chest`).
* **For Impacts**: Describe the collision, the reaction, and the result (`a heavy hammer strikes the glowing iron, sending a shower of bright orange sparks flying into the air; the metal visibly deforms under the blow`).
* **For Movement/Friction**: Detail the interaction between surfaces (`a car's tires screech, leaving black rubber marks on the asphalt as it drifts around a corner`).
* **For Natural Forces**: Describe the effect of the force on the environment (`trees bend and sway violently under the force of the wind, loose leaves are whipped into a frenzy`).
4. **Visual Style and Quality**:
* Specify the artistic style (`photorealistic`, `cinematic`), lighting (`dramatic, warm light from the forge`), and visual quality (`highly detailed, 4K`).

**Final Instruction**: Synthesize all these visual and physical elements into a single, rich, and coherent paragraph. Your entire output should be a prompt that visually and dynamically directs an AI. **Do NOT describe sound itself, but rather the physics of its creation.**

Based on the video, the prompt for video generation is:

Figure 5: Prompts for Stage1: Detailed Visual Scene Description

***Instruction for: Stage2: Auditory Concept Abstraction***

You are an expert AI assistant specializing in identifying auditory concepts from text. Your task is to read a descriptive video caption and extract a list of the primary objects or events that produce sound.

**Instructions:**

1.Analyze the provided caption to understand the scene.

2.Identify only the key elements that would create distinct sounds.

3.Ignore general scenery, static objects, or characters not directly involved in making a sound.

4.Output the concepts as a simple, comma-separated list of keywords.

**Example:**

**Input Caption:** A strong blacksmith forcefully strikes a hammer on glowing iron in the workshop, sending a shower of sparks.

**Audio Labels: Hammer, Sparks**

**Your Task:**

**Input Caption:** {video_caption}

**Audio Labels:**

Figure 6: Prompts for Stage2: Auditory Concept Abstraction

> **Instruction for Stage3: Visually-Grounded Audio Caption Generation**
>
> **TASK**
> Generate a concise, descriptive audio-only caption. You will be given a detailed video caption for context and a short list of key audio labels. This generated caption will be used as a prompt for a text-to-audio model, so it must describe a single, co-existing soundscape.
>
> **GUIDELINES**
> 1. Audio Only: The caption MUST describe only the audible aspects. Do not mention or allude to any visual information from the video caption.
> 2. No Sequences: DO NOT use temporal words (e.g., "first," "then," "starts with," "followed by"). Describe all sounds as if they are happening together in one scene.
> 3. Describe Characteristics: Based on the labels, describe the sound's quality.
> *Human Sounds: e.g., "a low-pitched voice," "a loud shout."
> * Sound Effects: e.g., "a door creaking," "a hammer striking metal."
> * Music: e.g., "an upbeat electronic drum beat," "a slow acoustic guitar."
> * Environment: e.g., "the rumble of city traffic," "wind howling," "birds chirping."
> 4. Final Output: The result must be a single, continuous audio caption that combines the elements from the audio labels into one coherent phrase.
> **EXAMPLE**
> Input Context:
> * Video Caption: A young man with a red backpack walks briskly down a paved path in a large city park during autumn. The wind is strong, whipping through the nearly bare trees and rustling his jacket. He wears large, black headphones, nodding his head to a beat. A small terrier on a leash ahead of him stops suddenly and lets out a sharp bark at a squirrel.
> * Audio Labels: `["Strong Wind", "Rock Music with Electric Guitar", "Dog Bark"]`
> Correct Output:
> * Audio Caption: The powerful roar of strong wind, a sharp dog bark, and the driving sound of rock music with an electric guitar all blend together in a dynamic soundscape.
> **YOUR TASK**
> Video Caption: {video_caption}
> Audio Labels: ["{audio_labels}"]
> Audio Caption:

Figure 7: Prompts for Stage3: Visually-Grounded Audio Caption Generation

## C.7  DETAILED COMMAND FOR HUMAN ANNOTATION

*Instruction for Human Annotation*

**Overview** This evaluation task consists of 150 samples. Each sample folder contains 7 videos that were generated based on the same text prompt but feature different audio and video outputs.

The videos are labeled 1 through 7. These are the outputs of 7 different models, and the order has been randomized for a blind evaluation. Please watch each video and rate it across the five dimensions below.

**Scoring Scale:** All dimensions are to be rated on a scale of **0 to 5**, using **0.5-point increments**.

**Evaluation Dimensions**

**1. Video Quality**

•**Scale:** 0-5

•**Focus:** Evaluate the visual quality of the video ONLY.

•**Criteria:** Assess the clarity, stability, and temporal consistency of the footage. Are there any distortions or artifacts? Does the video have natural, fluid motion, or does it appear static and unnatural (e.g., like a frozen frame or lagging)? The higher the visual fidelity, the higher the score.

**2. Audio Quality**

•**Scale:** 0-5

•**Focus:** Evaluate the audio track ONLY.

•**Criteria:** Assess the realism and clarity of the sound. Is the audio clean and free of noise or artifacts? The higher the audio fidelity, the higher the score.

**3. Text-AV Alignment (Audio & Video)**

•**Scale:** 0-5

•**Focus:** Evaluate how well the video and audio semantically match the provided text prompt.

•**Criteria:** Does the content of the video (what you see) and the audio (what you hear) accurately reflect the description in the text? The better the alignment, the higher the score.

**4. Audio-Video Synchronization**

•**Scale:** 0-5

•**Focus:** Evaluate the synchronization between the audio and video tracks.

•**Criteria:** Assess both temporal and semantic synchronization. For example, if a dog is shown barking, does the sound align with the visual of its mouth opening? Is the correct sound heard (e.g., a bark, not a meow)? Is there a noticeable delay? Does a sound event correspond to an appropriate visual change? The better the sync, the higher the score.

**5. Overall Impression**

•**Scale:** 0-5

•**Focus:** Provide a holistic rating of the video's overall quality.

•**Criteria:** Based on your comprehensive experience—considering the visuals, audio, alignment, and synchronization together—how compelling is the video as a whole? The better the overall viewing experience, the higher the score.

**Additional Notes:**

•**Sample 1** is provided as a labeled example to guide your ratings.

•**Important:** When rating each dimension, please focus exclusively on its specific criteria. Do not let your assessment of other dimensions influence your score for the current one.

Figure 8: Detailed Command for Human Annotation

## C.8 EXAMPLES RESULTS FOR HVGC FRAMEWORK

Table 6: Examples of captions generated by our HVGC Case1.

| | |
|---|---|
| **HVGC Video Caption** | In a dimly lit blacksmith workshop illuminated by the dramatic, warm light emanating from the forge, a weathered blacksmith with soot-stained hands stands at his anvil, his face etched with concentration. The atmosphere is thick with anticipation, the air filled with the scent of heated metal and the faint glow reflecting off the walls. The blacksmith, clad in a rugged leather apron, holds a glowing orange piece of iron with tongs, its intense heat casting flickering shadows across the room. He raises a heavy hammer above his head, muscles tensing as he prepares to strike. The hammer descends with force, colliding with the glowing iron and sending a shower of bright orange sparks flying into the air; the metal visibly deforms under the powerful blow. The blacksmith's arm recoils slightly from the impact, and he quickly adjusts his grip on the tongs, turning the iron to shape it further. Each strike is precise, the hammer connecting with the metal in a rhythmic dance of creation. The camera captures this intense moment from a close-up angle, focusing on the intricate details of the hammer's texture and the vibrant hues of the heated iron. The anvil, worn from years of use, bears the marks of countless creations, adding to the scene's authenticity. The visual counterpart of the hammer's resounding impacts is evident in the way the metal bends and reshapes, each deformation a testament to the blacksmith's skill. The entire scene is rendered in a photorealistic, cinematic style. |
| **HVGC Audio Caption** | The resounding clang of a heavy hammer striking metal echoes through the workshop, each powerful blow punctuated by the sizzle of heated iron and the occasional hiss of sparks. |
| **Omini Caption** | A blacksmith hammers a glowing, orange-hot metal blade on an anvil, shaping it with precise, rhythmic strikes. The intense heat radiates from the blade as sparks fly with each impact, while the sound of heavy hammering echoes in the workshop, capturing the raw power and craftsmanship of traditional metalworking. |
| **Audio-LLM_Caption** | There was a knocking sound, a bit like the sound of stones. |

Table 7: Examples of captions generated by our HVGC Case2.

| | |
|---|---|
| **HVGC Video Caption** | In a bustling subway station at Union Square, the atmosphere is alive with the vibrant energy of urban nightlife, illuminated by the stark white tiles and the soft glow of overhead fluorescent lights. A young man, dressed in a blue patterned shirt and dark shorts, stands as the focal point, his golden saxophone held close as he plays with fervent passion. His cheeks puff rhythmically, his fingers dance deftly across the instrument's keys, and his body sways to the beat, creating a visual counterpart to the music's lively tempo. Behind him, another musician, clad in black, sits with a large metallic drum, his hands moving in a blur as they strike the surface, sending vibrations through the air that ripple like invisible waves. The camera pans smoothly, capturing the dynamic interplay between the musicians and their instruments, emphasizing the physicality of their performance. Passersby weave through the scene, some pausing to listen, their presence adding layers of depth and movement. The walls are adorned with advertisements, one featuring a woman in casual attire, her image static yet contributing to the urban tapestry. The floor beneath the musicians' feet shows signs of wear, hinting at the constant flow of people and the passage of time. The visual style is crisp and photorealistic, with high contrast and sharp details that bring every element into vivid focus, creating a highly detailed, 8K representation of this captivating moment in the heart of the city. |
| **HVGC Audio Caption** | The rich, soulful notes of a saxophone fill the bustling subway station, blending with metal drum and the ambient hum of the urban environment. |
| **Omini Caption** | In the bustling underground corridor of Union Square, two street musicians ignite the space with raw energy — a saxophonist in a blue patterned shirt dances while blowing soulful notes from his golden baritone, and a drummer behind him pounds rhythmically on a snare strapped to his chest. Their music echoes off white-tiled walls as commuters blur past. |
| **Audio-LLM Caption** | A fast and cheerful piece of music, very unrestrained and passionate, like an orchestral piece. |

## D ABLATION STUDY ON BRIDGEDIT BLOCK PLACEMENT

Table 8: Ablation study on the placement of BridgeDiT Blocks. Performance is highest when interaction is focused on the early-to-mid layers of the architecture.

| Placement Strategy | Video Layers | Audio Layers | IB-VA ↑ | AV-Align ↑ |
|---|---|---|---|---|
| Early Layers | [0, 1, 2, 3] | [0, 1, 2, 3] | 28.30 | 0.2223 |
| Middle Layers | [13, 14, 15, 16] | [10, 11, 12, 13] | 31.89 | 0.2481 |
| Late Layers | [27, 28, 29, 30] | [21, 22, 23, 24] | 19.32 | 0.1831 |
| Uniform | [6, 12, 18, 24] | [2, 8, 13, 18] | 33.65 | 0.2502 |
| **Uniform (Early Bias)** | **[3, 7, 11, 15]** | **[2, 5, 8, 11]** | **34.59** | **0.2746** |

To understand the impact of the interaction module's placement, we conducted an ablation study by inserting four BridgeDiT Blocks at different stages within the dual-tower architecture. We evaluated five distinct placement strategies: concentrating the blocks in the early, middle, or late layers, as well as two uniform distribution strategies.

Table 9: Quantitative comparison on the AVsync-15 dataset of Figure 4. ↓ indicates lower is better, and ↑ indicates higher is better. The best results are highlighted in **bold**.

| Method | FVD↓ | FAD↓ | CLIPSIM↑ | CLAP↑ | VA-IB↑ | Av-Align↑ |
|--------|------|------|----------|-------|--------|-----------|
| Separate | 828.33 | 11.90 | 28.12 | 30.78 | 26.22 | 0.205 |
| Full-Fusion | 781.03 | 5.62 | 28.43 | 32.28 | 34.14 | 0.253 |
| Cross-V2A | 813.02 | 6.21 | 28.24 | 35.85 | 34.20 | 0.268 |
| Cross-A2V | **746.37** | 5.91 | 28.49 | 31.25 | 31.54 | 0.236 |
| Additive-Fusion | 772.23 | 5.72 | 28.20 | 34.77 | 28.34 | 0.258 |
| **DCA (Ours)** | 765.74 | **5.34** | **28.52** | **35.95** | **34.59** | **0.275** |

The results, presented in Table 8, reveal a clear trend. The Uniform (Early Bias) strategy, where blocks are inserted uniformly across the first half of the network layers, yields the best performance on both the ImageBind (IB-VA) and AV-Align metrics. Performance is strongest when interaction occurs in the early-to-mid layers, as seen in the "Middle Layers" and "Uniform" configurations. Conversely, concentrating the interaction exclusively in the deepest, final layers ("Late Layers") results in a significant degradation of performance. This suggests that for achieving robust audio-visual synchronization, the most critical feature exchange occurs at the early and intermediate representational stages. We hypothesize that these layers contain the optimal balance of detailed spatial-temporal information (from early layers) and abstract semantic concepts (from middle layers). Relying only on the highly abstract features from the final layers is insufficient for the precise alignment required for the T2SV task.

# E    ANALYSIS ON FUSION MECHANISMS

As shown in Table 9, these results conclusively demonstrate that our Dual Cross-Attention (DCA) mechanism consistently achieves superior or comparable performance across all evaluated metrics beyond audiovisual alignment. We clarify the structural necessity of our Dual Cross-Attention (DCA) compared to standard Full-Attention (FA). Let $L_V \in \mathbb{R}^{N_V \times D}$ and $L_A \in \mathbb{R}^{N_A \times D}$ denote video and audio latents.

- **Full-Attention (FA/Joint):** FA concatenates latents $L_{cat} = [L_V; L_A]$ and applies a single self-attention mechanism. The resulting attention matrix $M \in \mathbb{R}^{(N_V + N_A)^2}$ implicitly contains four sub-blocks:

$$M_{FA} = \begin{bmatrix} A_{V \to V} & A_{A \to V} \\ A_{V \to A} & A_{A \to A} \end{bmatrix}$$

  Critically, FA forces a single projection space to simultaneously model complex intra-modal dependencies ($V \to V$) and inter-modal alignment ($A \leftrightarrow V$).

- **Dual Cross-Attention (DCA):** DCA *never* concatenates. Instead, it employs two parallel, directed cross-attention layers to explicitly decouple interaction types. For the video stream (symmetrically for audio):

$$Q_V = L_V W_Q^V, \quad \{K_A, V_A\} = L_A \{W_K^A, W_V^A\}$$

$$L_V' = L_V + \text{Softmax}\left(\frac{Q_V K_A^T}{\sqrt{D_k}}\right) V_A$$

  Here, the attention map corresponds strictly to the $A \to V$ sub-block, isolating cross-modal injection from self-refinement.

Since BridgeDiT utilizes powerful pre-trained unimodal backbones (experts in intra-modal $A_{V \to V}/A_{A \to A}$), applying FA introduces redundancy by forcing the bridge to relearn intra-modal features. In contrast, DCA is **specialized**: it dedicates 100% of its parameters to bridging the modalities ($A \leftrightarrow V$). This targeted design avoids parameter waste and yields superior convergence, as evidenced by the performance gap in Table 3.

## F Limitation and Future Work

### F.1 Limitation

Despite the promising results, our work has several limitations. The primary challenge, shared by the entire T2SV field, is the scarcity of large-scale, high-quality, and well-annotated audio-video data. Our method's performance is highly dependent on data quality; unstable or low-resolution videos can degrade the capabilities of the pre-trained backbones, while noisy audio or the presence of out-of-frame sounds complicates the learning of precise synchronization. Our data filtering and hierarchical captioning are steps to mitigate this, but the need for better datasets remains. Furthermore, the current version of BridgeDiT is focused exclusively on generating sound effects. It does not yet support speech, which would require dedicated lip-synchronization mechanisms or complex musical scores. Finally, the overall performance of our model is inherently bounded by the capabilities of the chosen foundational T2V and T2A models, suboptimal base models significantly limit the overall generation quality.

### F.2 Future Work Direction

These limitations pave the way for several exciting future directions. A crucial step is the collection of larger, higher-quality audio-visual datasets, coupled with more efficient data processing pipelines for cleaning, filtering, and captioning. Building on our architecture, we plan to extend BridgeDiT to support speech and music. This will involve incorporating specialized modules for lip-synchronization and developing techniques to capture the rhythm and mood of musical inputs. Moreover, we are interested in exploring post-training refinement techniques. For instance, applying Reinforcement Learning with Human Feedback (RLHF), with rewards specifically designed to enhance audio-visual synchronization, could further improve the model's temporal and semantic coherence. We believe these future steps will continue to advance the field towards the generation of truly holistic and synchronized multi-sensory experiences.

