# OpenReview forum: "Taming Text-to-Sounding Video Generation via Advanced Modality Condition and Interaction"
_ICLR.cc/2026/Conference — Submitted to ICLR 2026_

### Official Review · Reviewer_yKFY · 2025-10-27

**Soundness:** 2
**Presentation:** 3
**Contribution:** 1
**Rating:** 2
**Confidence:** 4

**Summary:**

This paper addresses the problem of text-to-sounding video (videos with synchronized audio) generation. The proposed method initializes from a pretrained text-to-video network and a text-to-audio network. The main contributions are twofold:

1. A multistage, hierarchical prompt rewriting approach to generate video caption and audio caption separately instead of sharing one caption for both video and audio generation.
2. A “BridgeDiT” block that connects pretrained video and audio towers with attention layers.

**Strengths:**

This paper proposes a reasonable approach for text-to-sounding video generation. It is fairly simple, yet achieves strong performance compared with state-of-the-art models. The motivation for separating video and audio captions when pre-trained models are used makes sense.

**Weaknesses:**

My main concern is that the contributions are weak.

- Prompt rewriting is not new (has been used in Wan, for example). The proposed method uses multiple stages to rewrite the prompt, which brings additional complexity and is not carefully ablated. Table 3 compares the performance of a few captioning settings but does not ablate on the three stages. For instance, a single-stage method can also generate two types of captions for the video and the audio stream separately with in-context learning, which the proposed method has also used. I agree that using separate captions for video and audio makes sense when the network is initialized from pre-trained models, but this has a limited scope (the situation will be very different with a fully and jointly trained audio-video generation network) and limited contribution.
- If I am understanding correctly, the proposed “Dual Attention Fusion” module is a joint attention (“full-attention fusion” in the paper) with a masked block diagonal in the attention matrix. I don’t see why this is better than full attention, and I don’t see the justification for it in the paper. Can the authors elaborate? Figure 4 provides some training curves for these attention variants, but the curves don’t seem to have fully converged.

Moreover, I have some concerns about the experimental setup.

- Fairness of comparisons. Since the proposed method and some of the baselines (e.g., SSVG) bootstrap from pre-trained networks, the quality of these pre-trained networks plays a large role. This is evidenced by the results in Table 1 – the results (e.g., FVD and FAD) of the decoupled generation baselines are already better than the joint generation baselines. The proposed method uses these better pre-trained networks as backbones and compares with the prior works that use weaker pre-trained backbones. How do the authors ensure that the comparison is fair?
- Data leakage. From Section 4.1, it seems like the proposed method has been trained and evaluated on AVSync15, VGGSound-SS, and Landscape (though this is not clear; see question). It is known that AVSync15 and VGGSound have cross-contamination in the train/test split (MMAudio Section E). Have the authors considered this?

Wan: Open and Advanced Large-Scale Video Generative Models

**Questions:**

- Could the authors clarify the composition of the training set?
- Could this paper be compared with CoDi-2 on text-to-sounding video generation?
- Other questions in the weaknesses section.

---

> ### Author Response · Authors · 2025-11-22
> **Official Response to Reviewer yKFY [1/3]**
>
> **Thank you very much for taking the time to review and for your support. We try our best to address your questions as follows.**
>
> ## Q1: Critique of HVGC's Novelty, Complexity, and Ablation Analysis (Weakness 1)
> 1. > ...The proposed method uses multiple stages to rewrite the prompt...
>
> We would like to clarify the difference between our HVGC  framework and standard "prompt rewriting". The motivation of prompt rewriting is typically to enrich semantic expression for better instruction-following, as in Wan [1] and Seedance [2]. Our HVGC, in contrast, is designed specifically to solve the modal interference problem in dual-tower T2SV models. As we state, a single shared caption creates a conflict with the models' unimodal pre-training, which degrades performance. HVGC’s purpose is to perform modal disentanglement to resolve this conflict.
>
> 2. > Table 3 compares the performance of a few captioning settings but does not ablate on the three stages.
>
> Regarding the ablation of our three-stage process. Table 3 already demonstrates that HVGC outperforms the alternative disentangled baseline (using a direct Audio LLM). We agree that a more detailed ablation of the HVGC stages is beneficial. To address this, we have conducted new ablation experiments(results below). We compare HVGC against a single-stage baseline (as the reviewer suggested) using an Omini-modal model (Qwen2.5-Omini) with in-context learning to generate separate captions. We also ablate the necessity of Stage 2 by comparing against a variant that removes Stage 2. Due to time constraints, these new experiments are conducted in a zero-shot (no training) setting.
>
> | Method | FVD ↓ | FAD ↓| CLIPSIM ↑ | CLAP ↑ | VA-IB ↑ | AVAlign ↑ |
> | :--- | :--- | :--- | :--- | :--- | :--- | :--- |
> | Single-Stage (Omini-model) | 1423.48 | 16.89 | 25.37 | 23.86 | 23.89 | 0.157 |
> | HVGC (Ours, No Stage2) | **908.81** | 15.48 | 28.34 | 25.48 | 23.32 | 0.198 |
> | HVGC (Ours, Full 3-Stage) | **908.81** | **14.90** | **28.34** | **28.37** | **25.36** | **0.211** |
>
> As the results demonstrate, our full three-stage HVGC framework outperforms all simpler baselines. This confirms that our multi-stage design is not "additional complexity" but a necessary process to generate accurate, hallucination-free audio captions.
>
> 3. > ...  this has a limited scope (the situation will be very different with a fully and jointly trained audio-video generation network) and limited contribution...
>
> Regarding the reviewer's comment on the "limited scope" of our pre-trained paradigm. As discussed in our introduction, training a joint audio-video distribution from scratch (single-tower) remains data-intensive and computationally expensive, with current attempts showing limited success. In this context, the dual-tower approach, which efficiently leverages powerful pre-trained models, allows research to focus on the critical challenge of modal interaction. This Dual-tower paradigm presents a highly practical path forward for the open-source community, particularly at this stage of research.

---

> > ### Author Response · Authors · 2025-11-22
> > **Official Response to Reviewer yKFY [2/3]**
> >
> > ## Q2: Questions about DCA (Weakness 2)
> > >...I don’t see why this is better than full attention, and I don’t see the justification for it in the paper. Can the authors elaborate?...
> >
> > We thank the reviewer for the question and clarify the fundamental difference between our Dual Cross-Attention (DCA) and Full-Attention (FA).  We provide a theoretical analysis here.
> >
> > Let $L_V \in \mathbb{R}^{N_V \times D}$ and $L_A \in \mathbb{R}^{N_A \times D}$ be video and audio latents.
> >
> > **1. Full-Attention (FA)**
> >
> > FA concatenates latents, then applies standard Self-Attention:
> > $$
> > L_{cat} = \text{Concat}(L_V, L_A)
> > $$
> >
> > $$
> > Q_{cat} = \text{LN}(L_{cat})W_Q, \quad K_{cat} = \text{LN}(L_{cat})W_K, \quad V_{cat} = \text{LN}(L_{cat})W_V
> > $$
> >
> > The output of FA is:
> >
> > $$
> > L_{cat}^{'} = \mathrm{Softmax}\left(\frac{Q_{cat}K_{cat}^T}{\sqrt{D_k}}\right)V_{cat} + L_{cat}
> > $$
> >
> >
> > The attention matrix $A_{FA} \in \mathbb{R}^{(N_V+N_A) \times (N_V+N_A)}$ implicitly computes all four types of interactions:
> > $$
> > A_{FA} = \begin{bmatrix}
> > A_{V \to V} & A_{A \to V} \newline
> > A_{V \to A} & A_{A \to A}
> > \end{bmatrix}
> > $$
> > This structure forces a single module to learn both intra-modal ($A_{V \to V}, A_{A \to A}$) and inter-modal ($A_{A \to V}, A_{V \to A}$) relationships.
> >
> > ---
> >
> > **2. Our Dual Cross-Attention (DCA)**
> >
> > DCA uses two parallel, directed Cross-Attention operations. It **never concatenates** full latents.
> >
> > * **Audio-to-Video Cross-Attention:**
> >   Queries are from video, keys/values from audio.
> >   $$
> >   Q_V = \text{LN}(L_V)W_Q^V, \quad K_A = \text{LN}(L_A)W_K^A, \quad V_A = \text{LN}(L_A)W_V^A
> >   $$
> >   Output:
> >   $$
> >   \Delta L_V = \text{Softmax}\left(\frac{Q_V K_A^T}{\sqrt{D_k}}\right)V_A
> >   $$
> >
> >   $$
> >   L'_V = \Delta L_V + L_V
> >   $$
> >
> > * **Video-to-Audio Cross-Attention:**
> >   Queries are from audio, keys/values from video.
> >   $$
> >   Q_A = \text{LN}(L_A)W_Q^A, \quad K_V = \text{LN}(L_V)W_K^V, \quad V_V = \text{LN}(L_V)W_V^V
> >   $$
> >   Output:
> >   $$
> >   \Delta L_A = \text{Softmax}\left(\frac{Q_A K_V^T}{\sqrt{D_k}}\right)V_V
> >   $$
> >
> >   $$
> >   L'_A = \Delta L_A + L_A
> >   $$
> >
> >   Each attention matrix (e.g., $A_{V \leftarrow A}$) **only captures inter-modal relationships**. Distinct projection matrices ($W^V, W^A$) are used for each stream.
> >
> > Empirical results (Figure 4, Table 3) confirm DCA's superior performance, validating this specialised design.
> >
> > ----
> > > Figure 4 provides some training curves for these attention variants, but the curves don’t seem to have fully converged.
> >
> > Due to resource limitations, we only extended training to 15k steps. As shown in Figure 4, the curves have largely converged at this point, and we find that further training will degrade visual quality. Crucially, the relative performance ranking stabilised at 15k steps. We apologise for not being able to train for longer, and we plan to conduct further extended training on larger datasets in future work.

---

> ### Author Response · Authors · 2025-11-22
> **Official Response to Reviewer yKFY [3/3]**
>
> ## Q3: Fairness of comparisons (Weakness 3)
>
> We appreciate the reviewer's concern regarding the fairness of comparisons. While we acknowledge that superior backbones contribute to better quality, we respectfully disagree that our comparison is unfair.
>
> Firstly, we emphasize that our final setting compares against fine-tuned separated backbones. This comparison is fair in possessing the same pre-trained model and demonstrates the effectiveness of our joint module.
> Meanwhile, for single-tower baselines like CoDi, their performance reflects the inherent limitations of the "fully and jointly trained audio-video generation network" paradigm, which struggles with the extensive resources and data required for training from scratch. Their relatively lower performance is **a direct consequence of this architectural choice, not an unfair baseline.**
>
> Secondly, for dual-tower baselines, simply replacing their backbones with our own is highly costly. Many of these prior works do not provide open-source training code. It is also a common practice in the field to compare new methods against established baselines using their published configurations.
>
> Crucially, to directly address the concern of comparing fusion mechanisms under the same backbone, we have conducted an additional ablation study on various fusion strategies, all built upon identical backbones and trained with our HVGC captions. This ensures a fair and direct comparison of the interaction mechanisms themselves. We evaluated our DCA against several alternative fusion schemes:
>
> ●Full-Attention Fusion: Adopted from JointDiT [3].
>
> ●Cross-V2A: Similar to the video-to-audio conditioning in MTV [4].
>
> ●Cross-A2V: Similar to the audio-to-video conditioning in SyncFlow [5].
>
> ●Additive-Fusion: As utilized in SSVG [6].
>
> The comprehensive results, obtained under the same HVGC dataset training, are presented in the table below:
>
> | Model | FVD↓ | FAD↓ | CLIPSIM↑ | CLAP↑ | VA-IB↑ | Av-Align↑ |
> | :--- | :--- | :--- | :--- | :--- | :--- | :--- |
> | Separate | 828.33 | 11.90 | 28.12 | 30.78 | 26.22 | 0.205 |
> | Full-Fusion | 781.03 | 5.62 | 28.43 | 32.28 | 34.14 | 0.253 |
> | Cross-V2A | 813.02 | 6.21 | 28.24 | 35.85 | 34.20 | 0.268 |
> | Cross-A2V | **746.37** | 5.91 | 28.49 | 31.25 | 31.54 | 0.236 |
> | Additive-Fusion | 772.23 | 5.72 | 28.20 | 34.77 | 28.34 | 0.258 |
> | **DCA (Ours)** | $\underline{765.74}$ | **5.34** | **28.52** | **35.95** | **34.59** | **0.275** |
>
> These results conclusively demonstrate that our Dual-Cross Attention (DCA) has a leading performance. We will include the new table in the revised version.
>
> ## Q4: Question about training dataset (Weakness 4)
> We would like to clarify a crucial distinction: our work utilizes **VGGSound-SS [7], which is distinct from the broader VGGSound [8] dataset**. We have meticulously checked for overlaps between all datasets used in our study, namely AVSync15, VGGSound-SS, and Landscape. Our analysis confirms that there is no cross-contamination in the train/test splits across these datasets. We have performed a rigorous check for all splits using the same methodology as MMAudio, and the results are summarized in the following table:
> | Testset |  | |Train Set | |
> | :--- | :--- | :--- | :--- | :--- |
> | | | Avsync15 | VGGSound-SS | Landscape |
> | Avsync15 | - | 0.0% | 0.0% | |
> | VGGSound-SS | 0.0% | - | 0.0% | |
> | Landscape | 0.0% | 0.0% | - | |
>
> The Python script used for this leakage check, 	`caption_pipeline/recaption/check_leakage.py`, is included in our anonymous repo for verification.
>
> ## Q5: About the CoDi-2 baseline
>
> We appreciate the reviewer's suggestion to compare with CoDi-2. However, we would like to clarify that CoDi-2 is not designed for text-to-sounding video generation. It is primarily an MLLM-based model with an image encoder and integrated with Stable Diffusion for image generation. It does not possess the capability to generate temporally consistent sounding videos. Furthermore, CoDi-2 lacks well-supported open-source checkpoints and inference scripts (as can be seen at https://github.com/microsoft/i-Code/tree/main/CoDi-2 ), making a direct and fair comparison for our task practically infeasible at this time.
>
> [1] Wan: Open and Advanced Large-Scale Video Generative Models
>
> [2] Seedance 1.0: Exploring the Boundaries of Video Generation Models
>
> [3] Animate and sound an image. 2025 CVPR
>
> [4] Audio-Sync Video Generation with Multi-Stream Temporal Control
>
> [5] SyncFlow: Toward Temporally Aligned Joint Audio-Video Generation from Text
>
> [6] A Simple but Strong Baseline for Sounding Video Generation: Effective Adaptation of Audio and Video Diffusion Models for Joint Generation
>
> [7] VGGSound-SS: Localizing Visual Sounds the Hard Way
>
> [8] VGGSound: A Large-scale Audio-Visual Dataset"

---

> > ### Comment · Reviewer_yKFY · 2025-11-25
> >
> > I thank the authors for the response and the clarification. The added experiments and explanations on multi-stage ablations and the comparisons with different feature fusion strategies help clarify the contribution of this paper. The concerns about the experimental setups have been addressed. I'm raising my score to 4.
> >
> > For DCA, I now understand that it consists of two attention layers, and is thus substantially different than a single full joint attention layer. Are there any hypotheses as to why modeling only inter-modality relationships is better than modeling jointly? What are the implications on FLOPs (or speed) and parameter counts?

---

> ### Author Response · Authors · 2025-11-26
> **Official Response to Reviewer yKFY stage2 [1/2]**
>
> Dear Reviewer yKFY,
>
> Thank you for raising your score and for these excellent, fundamental questions. We are very happy to discuss with you.
>
> ## Q1：
> > Are there any hypotheses as to why modeling only inter-modality relationships is better than modeling jointly?
>
> We can explain our hypothesis from the perspective of the Manifold Assumption [1].
>
> Let's first consider the independent T2V and T2A generation. In this scenario, our video VAE and audio VAE are trained independently. We can therefore hypothesize that their latents, $z_v$ and $z_a$, exist on two distinct and uncorrelated latent manifolds, $M_v$ and $M_a$, respectively. Starting from a shared Gaussian prior $z \sim \mathcal{N}(0, I)$, the two generator branches ($G_\theta^V$ and $G_\theta^A$) learn to map this noise to their respective targets:
>
> $$
> G_\theta^V: z \to z_v, \quad \text{where } z_v \in M_v
> $$
>
> $$
> G_\theta^A: z \to z_a, \quad \text{where } z_a \in M_a
> $$
>
> When we aim for joint generation, our goal is to introduce an interaction module, or 'bridge' ($B_\theta^{AV}$), that connects these two branches. The full generative model is thus
>
> $$
> G_\text{model}= ( G_{\theta}^{A}, G_{\theta}^{V}, B_\theta^{AV} )
> $$
>
> Inside this bridge module, we must facilitate sufficient interaction. This is where the hypotheses for FA and DCA diverge.
>
> ### 1. Full-Attention (FA)
> From the Full-Attention perspective, we hypothesise that **FA works best by first "pulling"  the two distinct manifolds ($M_v, M_a$) into a single, unified joint manifold ($M_{joint}$)**. It attempts to use its QKV projection matrices to learn a shared representation *before* meaningful interaction can occur. This is a "merge-then-interact" approach:
>
> $$
> \text{FA path: } (L_v, L_a) \xrightarrow{\text{Project}} L_{joint} \xrightarrow{\text{Self-Attention}} L'_{joint} \to (L'_v, L'_a)
> $$
>
> This "manifold alignment" requires significant parameter capacity. In massive models like MMDiT [2] or SD3, which are trained with a vast number of parameters, FA is given enough power to successfully learn this complex $M_{joint}$ representation.
>
> ### 2. FA limitation
> However, our paradigm is different. As discussed in the paper, we operate under two crucial constraints:
>
> 1.  We use the dual-stream paradigm to leverage powerful pre-trained models.
> 2.  We must keep the backbone parameters **largely frozen** to retain this pre-trained knowledge.
>
> In this constrained scenario, the backbones themselves insist on operating within their original, separate manifolds ($M_v$ and $M_a$). FA is not given sufficient trainable capacity to "pull" these two powerful, separate streams into one. It struggles because its prerequisite (the ability to create a unified manifold) is not met.
>
> ### 3. Our DCA
> Our **Dual Cross-Attention (DCA)**, by contrast, is architecturally designed for this *exact* scenario. It does not need to align the manifolds. It operates as a lightweight "bridge" that facilitates direct, specialized interaction *between* the two independent streams:
>
> $$
> \text{DCA path: } (L_v, L_a) \xrightarrow{\text{Cross-Attn}} (L_v \leftarrow L_a, \ L_a \leftarrow L_v) \to (L'_v, L'_a)
> $$
>
>
> In this situation, DCA is tasked with learning the *mapping between* the manifolds ($M_v \leftrightarrow M_a$), which is a much more direct and well-defined task than FA's goal of creating an entirely new joint manifold ($M_{joint}$).
>
> ### 4. Q1 Summary
> In summary, because we chose the dual-stream, frozen-backbone paradigm (to leverage pre-training) and are thus limited in our parameter budget for interaction, FA cannot work effectively. **DCA provides a superior solution because it is specialised for interacting between separate manifolds.**
>
> Notably, we are not claiming that FA is universally inferior to DCA. We hypothesise that for a large-scale, AV-joint model trained from scratch with sufficient data and resources (which is hard), FA might be the superior choice. However, in the pre-trained, dual-tower paradigm, both our theory and experiments demonstrate that DCA is the more effective architecture.

---

> > ### Author Response · Authors · 2025-11-26
> > **Official Response to Reviewer yKFY stage2 [2/2]**
> >
> > ## Q2:
> > >  What are the implications on FLOPs (or speed) and parameter counts?
> >
> > We have uploaded our implementation to the anonymous repository (see `bridgedit/models/dit/dit_dual_stream_clean.py` ), which defines `FullDiTBlockAdaLN` (for FA) and `CrossDiTBlockAdaLN` (for DCA).
> >
> > ### 1. FLOPS
> >
> > Theoretically, Full-Attention cost is  $O((N_V+N_A)^2 D)$ while DCA is only $O(2 N_V N_A D)$.
> >
> > In practical Calculation,  using our training variables (`B=2`, `N_video=32,760`, `N_audio=1,025`, `D=1536`), the FLOPs for a single block are:
> >
> > | Component                                   | Full-Attention (FA)  | Dual Cross-Attn (DCA) |
> > | :------------------------------------------ | :------------------- | :-------------------- |
> > | **Base FLOPs** (MLPs, Norms, etc.)          | 3827.38 G-FLOPs      | 3827.38 G-FLOPs       |
> > | **Attention FLOPs** (The *only* difference) | 14025.85 G-FLOPs | **825.24 G-FLOPs**    |
> > | **Total (per Block)**                       | 17853.23 G-FLOPs | **4652.62 G-FLOPs**   |
> >
> >
> > ###  2. Parameter
> >
> > The parameter counts for our FA and DCA blocks are identical.
> >
> > - Full-Attention Block: ~84.9M Params
> >
> > - DCA Block: ~84.9M Params
> >
> > As shown in the code, both modules are constructed from the exact same set of components:
> >
> > The same 2 `MLP` blocks, 2 `adaln_modulation layers`, 4 `LayerNorm` layers, 8 QKV/O projection matrices `(q/k/v/o_proj_1 and q/k/v/o_proj_2)`.
> >
> > The difference is not in the parameters but in the `forward() `pass. FA concatenates all QKVs for one large, slow attention, while DCA uses the same QKVs for two separate, efficient cross-attentions.
> >
> > We believe this thorough discussion has been very valuable and will be a positive contribution to the community.
> >
> > Please let us know if you have any further questions. We are happy to discuss. : )
> >
> > Best regards,
> >
> > [1]  Back to Basics: Let Denoising Generative Models Denoise
> >
> > [2] Scaling Rectified Flow Transformers for High-Resolution Image Synthesis

---

### Official Review · Reviewer_K86T · 2025-10-27

**Soundness:** 1
**Presentation:** 3
**Contribution:** 2
**Rating:** 2
**Confidence:** 4

**Summary:**

This paper introduces a new framework for generating captions for sounding videos and proposes a new model architecture for sounding video generation. The captioning framework provides modality-specific captions for both audio and video. It first creates a detailed video caption from a given video clip using VLLM. Then, another LLM extracts sounding events from the video caption and creates a corresponding audio caption. These captions are used to train a new model called BridgeDiT, which is constructed by combining two pretrained DiT models for audio and video through dual cross-attention fusion. The experimental results show that the proposed model outperforms several baselines, including independent generation methods, sequential generation methods, and joint generation methods.

**Strengths:**

- This work addresses a challenging and important problem: the efficient construction of sounding video generation models. While numerous video and audio generation models have been released, models capable of jointly generating audio and video remain limited. Efficiently constructing such models is a hot topic in this field.
- The design of the proposed model is simple and applicable to any DiT-based models, which are a standard choice for recent audio and video generation tasks.
- The idea of creating dedicated text prompts for audio and video seems generally reasonable, though there are concerns about how to create them, which I will mention in the Weaknesses section.

**Weaknesses:**

- The empirical findings obtained through this work are somewhat limited, and it is not clear if they are sufficiently general.
  - Since the proposed architecture is widely applicable to DiT-based models, it is highly recommended to examine it with various backbones other than Wan + Stable Audio Open.
  - Several empirical findings are interesting; for example, a few BridgeDiT blocks are enough to construct sounding video generation models, and dual cross-attention is more effective than full-attention or addition. However, it is not clear if these results depend on the choice of backbone models.
- It would be helpful if the authors could show how to ensure that the generated audio caption accurately describes the content in the soundtrack of the source sounding video.
  - Since the proposed captioning framework solely relies on visual content in a sounding video, the generated audio caption is essentially limited to on-screen audible events. This means that any off-screen sounds could degrade the quality of the audio caption, making the training data noisier.
  - The datasets used in the experiments are all carefully filtered, so I think this may not be a significant issue in the experiments. However, it would be highly desirable in practice to have a validation stage in the framework to ensure that the generated audio caption aligns with the actual audio of the sounding video.
- Regarding the proposed captioning framework, it is not clearly described how spatial information is handled.
  - Since this study uses Stable Audio Open as a backbone for audio generation, the proposed model can generate stereo audio.
  - Therefore, spatial information about audible objects in a scene is important for the corresponding caption to properly describe the audio content.
  - However, this aspect is not mentioned in the instruction prompts shown in the appendix.
- The demo page should include results from baselines for qualitative comparison.

**Questions:**

- Did the authors examine the proposed architecture with other backbones?
- How can we ensure that the generated audio caption accurately describes the content in the soundtrack of the source sounding video?
- How is spatial information handled in the proposed captioning framework?

---

> ### Author Response · Authors · 2025-11-22
> **Official Response to Reviewer K86T**
>
> **Thank you very much for taking the time to review and for your support. We try our best to address your questions as follows.**
> ## Q1:  Proposed architecture with other backbones
> We recognize the importance of validating our approach beyond our initial Wan and Stable Audio Open backbones. Therefore, we conducted additional experiments using SANA Video [1] (a fast, linear DiT for video) and AudioLDM [2] (a UNet-based audio model), trained for 7500 steps due to time constraints. The results below demonstrate our method's generalizability:
> | Model Name | FVD↓ | FAD↓ | CLIP Sim↑ | CLAP↑ | Imagebind↑ | AVAlign↑ |
> | :--- | :--- | :--- | :--- | :--- | :--- | :--- |
> | SANA + AudioLDM (Separate) | 1033.39 | 17.09 | 27.95 | 29.66 | 23.39 | 0.193 |
> | SANA + AudioLDM (FullAttention) | 992.83 | 11.51 | 28.35 | 30.73 | 28.69 | 0.221 |
> | SANA + AudioLDM (DCA) | **928.74** | **10.34** | **28.92** | **32.67** | **30.23** | **0.242** |
>
> Even with limited training, our method using Dual CrossAttention (DCA) consistently outperforms the separate baseline with the new backbones. DCA remains more effective than Full-Attention for inter-modal fusion, confirming the generalizability of our approach across different architectures.
>
> ## Q2: Ensuring Alignment of Generated Audio Captions with Actual Audio Content
> We thank the reviewer for this insightful question, which highlights a crucial aspect of visually-grounded audio captioning. We acknowledge that our HVGC framework is intentionally designed to focus on on-screen audible events, as strong visual-audio correlation is paramount for coherent T2SV generation. Off-screen sounds often introduce ambiguity and noise, hindering this goal.
>
> To ensure our captions accurately describe relevant audio content and remove noise from off-screen sounds, we rely on a data filtration process rather than a post-hoc validation stage. This process involves:
>
> 1. Filtering out videos labeled with "talking," "speech," or background music.
>
> 2. Removing videos with low audio-visual alignment (ImageBind score < 0.3).
>
> Crucially, to quantitatively validate that our generated captions align with the actual soundtrack, we evaluated the audio-text alignment using the CLAP score on the final AVSync15 dataset. We compared our HVGC captions against the original dataset captions and those from a baseline Audio LLM.
> | Caption Source | CLAP Score (Audio-Text Alignment) ↑ |
> | :--- | :--- |
> | Omini Captions | 0.354 |
> | Audio LLM Captions | 0.243 |
> | **HVGC Captions (Ours)** | **0.384** |
>
> As shown in the table, our HVGC captions achieve the highest CLAP score, indicating a superior alignment with the actual audio content compared to both original and Audio LLM-generated captions. This quantitative evidence demonstrates that our filtration and visually-grounded generation process effectively ensures that the resulting captions accurately describe the relevant soundtrack content.
>
> ## Q3:  About spatial information in the Audio
> We agree that sounding video with stereo audio is a highly promising direction. However, current sounding video generation models, including Veo 3 and Sora 2, primarily focus on mono audio. Furthermore, existing academic datasets are currently mono-only and do not support stereo audio. Therefore, in this paper, we exclusively consider mono audio generation. When utilizing Stable Audio Open, we handle this by duplicating the single audio track for stereo output. We appreciate the reminder and will clarify this detail in the appendix of the revised version.
>
> ## Q4: About More demos on the demo page
> Thank you for the reminder. We have added more examples of other baselines to the demo page. https://bridgedit-t2sv.github.io/
>
> [1] SANA-Video: Efficient Video Generation with Block Linear Diffusion Transformer
>
> [2] Audioldm: Text-to-Audio Generation with Latent Diffusion Models

---

> > ### Comment · Reviewer_K86T · 2025-11-25
> >
> > Q1:
> >
> > Thanks for conducting the additional experiments. I will respond to this point once all results are provided here.
> >
> > Q2:
> >
> > Thanks for the clarification. I have two follow-up questions:
> > - What do the authors mean by "labeled" at the first step in the data filtering process? Is it assumed that the source video has some annotation?
> > - Do the authors think that CLAP score can accurately ensure the audio-text alignment? If so, why not use it for post-hoc validation?
> >
> > Q3:
> >
> > Thanks for the clarification. I do not have any further question on this point.
> >
> > Q4:
> >
> > Thanks for updating the demo page. It would be much helpful if the authors also provide some examples used in the user study (Section 4.5).

---

> ### Author Response · Authors · 2025-11-25
> **Official Response to Reviewer K86T**
>
> Dear Reviewer K86T,
>
> Thank you for engaging in this discussion. We are happy to provide further clarifications.
> ## Q1:
> We have now completed the additional experiments, training for 15k steps (consistent with our original setup). Here are the complete results:
>
> | Model Name                      | FVD↓       | FAD↓     | CLIP Sim↑ | CLAP↑     | Imagebind↑ | AVAlign↑  |
> | :------------------------------ | :--------- | :------- | :-------- | :-------- | :--------- | :-------- |
> | SANA + AudioLDM (Separate)      | 1033.39    | 17.09    | 27.95     | 29.66     | 23.39      | 0.193     |
> | SANA + AudioLDM (FullAttention) | 954.34     | 10.28    | 28.42     | 31.65     | 29.47      | 0.234     |
> | SANA + AudioLDM (DCA)           | **906.29** | **9.56** | **28.86** | **33.12** | **31.85**  | **0.256** |
>
> This demonstrates that our DCA is effective for diverse architectures, including Linear-Attention DiT (SANA) and UNet-based models (AudioLDM).
>
> Furthermore, these new backbones (SANA + AudioLDM) revalidate the superiority of DCA over Full-Attention. As we also detailed in `Official Response to Reviewer yKFY [2/3]`, this is because Full-Attention forces a single module to learn both intra-modal and inter-modal relationships. In contrast, our DCA is designed to only capture inter-modal relationships.
>
> ## Q2:
> - You are correct. By "labeled," we refer to the coarse-grained annotations that are already provided with datasets like VGGSound-SS. We use these existing labels for a preliminary filtering step to remove videos containing "talking," "speech," etc.
> - On CLAP as post-hoc validation: This is a very insightful question. We view CLAP as a useful indicator of alignment, but we do not believe it is fully reliable on its own to be used as a post-hoc filter. We believe **the most direct and trustworthy metric for a caption's quality is the final generation performance** (e.g., FVD, FAD) of the model trained on those captions. Therefore, in our main paper, we reported the final generation metrics as the primary evaluation of caption quality in Table 3. The CLAP score, which we added in the rebuttal, serves as a helpful auxiliary metric to analyze the intermediate caption alignment.
>
> ## Q4:
> Thank you for the reminder. We have updated our demo page to include some examples used in the user study (Section 4.5) as you requested.
>
> Please let us know if you have any further questions. We are happy to discuss. : )
>
> Best regards,

---

> > ### Comment · Reviewer_K86T · 2025-11-27
> >
> > A1:
> >
> > Thank you for providing the additional results.
> > - Are these results obtained on the AVSync15 dataset?
> > - Does the proposed model here also follow the early-bias strategy to determine the placement of BridgeDiT blocks?
> >
> > A2:
> >
> > Thank you for the clarifications.
> > Just to confirm, is the main reason for using coarse-grained annotations the difficulty of detecting off-screen sounds solely from visual cues?
> > - My understanding is that, since ImageBind only evaluates semantic alignment, filtering out video clips with off-screen sounds relies on these coarse-grained annotations. I would like to make clear what requirements the dataset should meet.
> > - Also, as you mentioned, CLAP is not sufficiently reliable for this filtering purpose.

---

> > > ### Author Response · Authors · 2025-11-27
> > > **Official Response to Reviewer K86T**
> > >
> > > Dear Reviewer K86T，
> > >
> > > Thank you for engaging in this discussion. We are happy to clarify these final details.
> > >
> > > ## For A1：
> > >
> > > - Dataset：Yes. New experimental results are trained on the AVSync15 dataset.
> > > - BridgeDiT Placement Strategy: Yes. We followed the exact same "Early Bias" strategy identified as optimal in our paper (Appendix D, Table 8). We believe these layers contain the optimal balance of detailed spatial-temporal information (from early layers) and abstract semantic concepts (from middle layers).  As our new backbones have different depths (SANA-video has 20 layers, AudioLDM has 16), we adapted the placement proportionally (e.g., placing the 4 blocks at layers `[ 3, 5, 7, 9]` for SANA and `[2, 4, 6, 8]` for AudioLDM).
> > >
> > > ## For A2:
> > > This is an important clarification, as the coarse-grained labels and ImageBind actually **serve two independent roles** in our filtering.
> > >
> > > Our primary task is focused on sound effects, not including speech. This is mainly because our backbone (Stable Audio Open) cannot generate speech, as open-source models that jointly generate both high-quality sound and speech (like Veo3 or Sora2) are not yet available. Therefore, we must filter out speech-heavy data. The coarse-grained labels are simply an effective way for us to filter out these scenarios (e.g., "talking").
> > >
> > > We use ImageBind to filter for general audio-visual semantic alignment. This step helps remove videos with poor correlation, which includes noise from significant off-screen sounds.
> > >
> > > Regarding data requirements,  our data requirements are not overly strict. We just need sounding video data that, after our filtering, contains no speech and has good audio-visual alignment. General datasets can largely meet this. If an open-source Text-to-Sound-and-Speech model becomes available, we could easily adapt our method to include speech.
> > >
> > > Please let us know if you have any further questions. We are happy to discuss. : )
> > >
> > > Best regards,

---

### Official Review · Reviewer_WTCT · 2025-10-29

**Soundness:** 3
**Presentation:** 3
**Contribution:** 3
**Rating:** 6
**Confidence:** 4

**Summary:**

This paper proposes a Text-to-Sounding-Video (T2SV) generation framework that integrates two key components: Hierarchical Visual-Grounded Captioning (HVGC) for generating modality-specific captions for both video and audio, and Dual CrossAttention (DCA) fusion mechanism for optimal audiovisual interaction.
Experimental results show that the proposed method achieves consistent improvements over existing baselines, including sequential generation (T -> V -> A or T -> A -> V) and joint generation models.

**Strengths:**

1. The proposed model is conceptually simple yet effective. DCA modules are inserted into only four layers, which likely achieves strong audiovisual alignment with modest computational cost.
2. HVGC addresses a critical but underexplored issue: how to construct suitable captions for each modality in sounding video generation.
3. Experiments on multiple benchmarks demonstrate clear and consistent gains. The paper includes well-chosen baselines, including sequential and joint generation pipelines, providing fair and meaningful comparisons.

**Weaknesses:**

1. The experimental setup lacks clarity. It is unclear whether models were trained separately for each dataset or jointly across all datasets. More importantly, it is unclear whether baseline models were trained with the same HVGC captions or standard captions, which directly affects the fairness of the comparison.
2. The analysis of fusion mechanisms is limited to audiovisual alignment. Although the reviewer understands that audiovisual alignment is the most important metric for DCA, video and audio quality, and text alignment are also crucial. Further comparison of DCA with alternative fusion schemes would strengthen the analysis.
3. The user study involves only five participants, which limits the statistical significance of the subjective evaluation.

**Questions:**

1. Did the authors train separate models for each dataset or a single unified model?
2. Were baseline models trained or fine-tuned on the same dataset using HVGC captions?
3. Have the authors evaluated DCA against other fusion mechanisms beyond audiovisual alignment?

---

> ### Author Response · Authors · 2025-11-22
> **Official Response to Reviewer WTCT**
>
> **Thank you very much for taking the time to review and for your support. We try our best to address your questions as follows.**
>
> ## Q1: Clarification of Experimental Setup.
> We appreciate the reviewer's request for clarification regarding our experimental setup. Below, we address the questions about model training and baseline comparisons.
> 1. > ...It is unclear whether models were trained separately for each dataset or jointly across all datasets ...
>
> We trained our models separately for each dataset. This approach was chosen because each dataset serves as a distinct benchmark. By training separate models, we can evaluate our method's performance on each specific benchmark, thereby demonstrating its effectiveness across diverse scenarios rather than aiming for a single general-purpose model.
>
> 2. >... it is unclear whether baseline models were trained with the same HVGC captions or standard captions, which directly affects the fairness of the comparison...
>
> To ensure a fair comparison, we categorized the baselines into two types:
>
> ● Pre-trained Models (CoDi, JavisDiT): We utilized the publicly available pre-trained models for inference without additional training. These models were already trained on extensive datasets, such as VGGSound and Landscape, making further training on our specific benchmarks redundant.
>
> ● Constructed Baselines: For the baselines we constructed, all models were trained using our HVGC captions. This ensures consistency in captioning across these baselines, allowing for a fair evaluation of the performance gains attributable to our proposed architecture.
>
> ## Q2: Other fusion mechanisms beyond audiovisual alignment
> We thank the reviewer for the insightful suggestion to broaden our analysis of fusion mechanisms. We agree that a comprehensive evaluation should include video quality, audio quality, and text alignment in addition to audiovisual alignment. We have conducted further evaluations on the AVSync15 dataset, and the results are presented below:
>
> | Model | FVD ↓ | FAD ↓ | CLIPSIM ↑ | CLAP ↑ | VA-IB ↑ | Av-Align ↑ |
> | :--- | :--- | :--- | :--- | :--- | :--- | :--- |
> | Separate | 828.33 | 11.90 | 28.12 | 30.78 | 26.22 | 0.205 |
> | Full-Fusion | 781.03 | 5.62 | 28.43 | 32.28 | 34.14 | 0.253 |
> | Cross-V2A | 813.02 | 6.21 | 28.24 | 35.85 | 34.20 | 0.268 |
> | Cross-A2V | **746.37** | 5.91 | 28.49 | 31.25 | 31.54 | 0.236 |
> | Additive-Fusion | 772.23 | 5.72 | 28.20 | 34.77 | 28.34 | 0.258 |
> | **DCA (Ours)** | $\underline{765.74}$ | **5.34** | **28.52** | **35.95** | **34.59** | **0.275** |
>
> Our Dual Cross-Attention (DCA) mechanism consistently achieves superior or comparable performance across all evaluated metrics beyond audiovisual alignment.
>
> ## Q3: About More User Study
> We thank the reviewer for this constructive feedback regarding the statistical significance of our user study. To address this valid concern, we have expanded our subjective evaluation by commissioning a professional annotation company to recruit an additional 15 participants, bringing the total number of evaluators to 20.
> | Method | VQ↑| AQ↑| TA↑ | Sync↑ | Overall↑|
> | :--- | :---: | :---: | :---: | :---: | :---: |
> | Wan+SDA | 3.16 | 2.93 | 2.75 | 2.47 | 2.79 |
> | Wan+Seeing | 3.16 | 2.83 | 2.91 | 2.54 | 2.85 |
> | Wan+MMAudio | 3.16 | 3.18 | 3.07 | 2.77 | 3.06 |
> | SDA+TmToken | 1.97 | 2.36 | 2.00 | 1.83 | 2.04 |
> | JointDiT | 2.74 | 2.81 | 2.79 | 2.45 | 2.69 |
> | JavisDiT | 2.47 | 2.39 | 2.36 | 2.14 | 2.36 |
> | **Bridgedit(ours)** | **3.40** | **3.33** | **3.33** | **3.09** | **3.34** |
>
> The updated results confirm that our method consistently outperforms baselines across all metrics, aligning with the initial study's conclusions. We will update these results in the revised paper.

---

> > ### Comment · Reviewer_WTCT · 2025-11-25
> >
> > Thank you for your work. My concerns regarding Q2 and Q3 have been addressed.
> >
> > However, Q1 is still not adequately addressed. In particular, the comparison between pre-trained models without any additional training or fine-tuning and BridgeDiT remains unfair. According to Table 3, training on HVGC captions substantially improves overall performance. The proposed dataset construction pipeline uses LLM-generated captions, which differ significantly from those used in VGGSound (short phrases) and Landscape (scene labels). This creates a clear domain gap between what the pre-trained models were trained to expect and the captions actually used at inference time. Therefore, training the pre-trained models on HVGC captions is still necessary for a fair comparison, regardless of how many videos were used during their original training.
> >
> > In short, the current evaluation does not convincingly demonstrate the effectiveness of DCA. It remains unclear whether DCA truly captures audiovisual interactions better than other designs, or whether the model overfits to the HVGC captions. To fairly assess the effectiveness of the DCA module, pre-trained models (e.g., JavisDiT) should also be trained on the same caption set. Alternatively, could the authors provide a more detailed explanation of why this caption-level domain gap would not affect the base models?

---

> > > ### Author Response · Authors · 2025-11-25
> > > **Official Response to Reviewer WTCT**
> > >
> > > Dear Reviewer WTCT,
> > >
> > > Thank you for this excellent follow-up question. This is a crucial point, and we appreciate the chance to clarify our evaluation strategy.
> > >
> > > ## 1. Some Clarifications
> > >
> > > > The proposed dataset construction pipeline uses LLM-generated captions... This creates a clear domain gap...
> > >
> > > We would like to clarify this point that most T2SV baselines do not train on VGGSound (short phrases) nor Landscape (scene labels) neither. They each use their own methods to generate text conditions. **In general, most text conditions are detailed, natural language descriptions of the video**, generated by LLMs. For example, Uniform [1] uses PLlaVA [2], SyncFlow [3] uses VideoOFA [4], and JavisDiT [5] uses Qwen2-VL-72B. Since our HVGC captions fall into this same "detailed natural language" domain, **we acknowledge this comparison is not perfectly ideal. However, we believe the Out-of-Distribution (OOD) problem may not be as significant**, as our inference conditions are also detailed natural language descriptions of the video.
> > >
> > > > To fairly assess the effectiveness of the DCA module, pre-trained models should also be trained on the same caption set.
> > >
> > > We agree that re-training all baselines on HVGC would be ideal. However, this is unfortunately infeasible for two main reasons: (1) Many models only release their inference code, not their training code. Some are not open-sourced at all, making fine-tuning impossible. (2) Many of these models accept only a single one-modal text condition, making their architectures incompatible with our data format.
> > >
> > > Despite these challenges, we have still made significant efforts to conduct a fair comparison
> > >
> > > ## 2. Our Effect on fair comparison with our DCA fusion
> > >
> > >  ### (1) Same backbone and training dataset with different methods
> > > In this work, we actually 、some baselines with identical backbones and train them with our identical HVGC captions. This ensures a fair and direct comparison of the interaction mechanism itself.
> > > We evaluated our DCA against alternative schemes:
> > >
> > > - Full-Attention Fusion (Adopted from JointDiT [6])
> > >
> > > - Cross-V2A (Adopted from MTV [7])
> > >
> > > - Cross-A2V (Adopted from SyncFlow [3])
> > >
> > > - Additive-Fusion (As utilized in SSVG [8])
> > >
> > > The full results (in our `official Response to Reviewer WTCT` Q2 response table) show that our DCA outperforms all other methods in this fair, controlled setting. This confirms the benefit is from the architecture, not just the data.
> > >
> > > ### (2) Validation on New Backbones
> > > To prove this is not specific to our original backbone, we ran the same test (DCA vs. Full-Attention) on entirely new architectures: SANA Video [9] (linear DiT) and AudioLDM [10] (UNet).
> > > | Model Name                      | FVD↓       | FAD↓     | CLIP Sim↑ | CLAP↑     | Imagebind↑ | AVAlign↑  |
> > > | :------------------------------ | :--------- | :------- | :-------- | :-------- | :--------- | :-------- |
> > > | SANA + AudioLDM (Separate)      | 1033.39    | 17.09    | 27.95     | 29.66     | 23.39      | 0.193     |
> > > | SANA + AudioLDM (FullAttention) | 954.34     | 10.28    | 28.42     | 31.65     | 29.47      | 0.234     |
> > > | SANA + AudioLDM (DCA)           | **906.29** | **9.56** | **28.86** | **33.12** | **31.85**  | **0.256** |
> > >
> > > This demonstrates that our DCA is effective for diverse architectures, including Linear-Attention DiT (SANA) and UNet-based models (AudioLDM).
> > >
> > > ### (3) Theoretical Justification
> > > As detailed in `Official Response to Reviewer yKFY [2/3]`, DCA is theoretically more specialized. Full-Attention forces a single module to learn both intra-modal and inter-modal relationships, while our DCA is designed to only capture inter-modal relationships.
> > >
> > > ### In summary, while we acknowledge that we do not perform a perfect comparison with T2SV baselines due to existing challenges. We are trying our best to make the comparison fair.
> > >
> > > Please let us know if you have any further questions. We are happy to discuss. : )
> > >
> > > Best regards,
> > > Authors.
> > >
> > > [1] UniForm: A Unified Diffusion Transformer for Audio-Video Generation
> > >
> > > [2] PLLaVA : Parameter-free LLaVA Extension from Images to Videos for Video Dense Captioning
> > >
> > > [3] SyncFlow: Toward Temporally Aligned Joint Audio-Video Generation from Text
> > >
> > > [4] VideoOFA: Two-Stage Pre-Training for Video-to-Text Generation
> > >
> > > [5] Animate and sound an image. 2025 CVPR
> > >
> > > [6] Audio-Sync Video Generation with Multi-Stream Temporal Control
> > >
> > > [7] A Simple but Strong Baseline for Sounding Video Generation: Effective Adaptation of Audio and Video Diffusion Models for Joint Generation
> > >
> > > [8] SANA-Video: Efficient Video Generation with Block Linear Diffusion Transformer
> > >
> > > [9] Audioldm: Text-to-Audio Generation with Latent Diffusion Models

---

> ### Comment · Reviewer_WTCT · 2025-11-25
>
> Thank you for clarifying the text prompts used in prior work. I also appreciate the authors’ efforts to ensure a fair comparison.
> Although the explanation is appreciated, it does not provide sufficient justification for me to revise my recommendation significantly. I will therefore maintain my score at 6. I look forward to further discussion of this submission with the other reviewers.

---

> ### Author Response · Authors · 2025-11-25
> **To Reviewer WTCT**
>
> Dear Reviewer WTCT,
>
> Thank you very much for your time and for actively engaging in this discussion. We truly appreciate your insightful feedback.
>
> We believe this thorough discussion has been very valuable and will be a positive contribution to the community.
>
> We wish you a nice day : ) !

---

### Official Review · Reviewer_SK8e · 2025-11-03

**Soundness:** 3
**Presentation:** 3
**Contribution:** 3
**Rating:** 2
**Confidence:** 3

**Summary:**

This papers explores the text-to-sounding-video generation task, in the context of jointly fine-tuning a pair of pretrained text-to-audio and  text-to-video generation models, with the following contributions:
- Use different text prompts for each tower as opposed to a single prompt in previous works.
- A Hierarchical Visual-Grounded Captioning (HVGC) framework meant to generate pairs of disentangled captions: one for video with emphasis on the visual aspect, one for audio with an emphasis on sound.
- Incorporate a dual cross attention mechanism in the joint model architecture for modality interaction, named BridgeDiT.

**Strengths:**

I believe that despite its apparent simplicity this paper is significant for the joint audio-video generation task.

Using different prompts for audio and video is clever and dual cross attention is a natural intermodal network design. This paper conducts rigorous ablations on both aspects to demonstrate their benefit for the task.

Moreover the HVGC builds on the intuition that audio captioning is more accurate when being in context of the visual scene.

**Weaknesses:**

The major weakness of the paper is the lack of novelty.
- Using different prompts for audio and video is clever but common sense.
- The visually grounded audio captioning tasks lack a related works section.
- Dual cross-attention architectures have been employed in the past for other multimodal tasks, yet this paper cites almost none of them (e.g. `Generative Spoken Dialogue Language Modeling` by Nguyen et al., `Dual-Stream Diffusion Net for Text-to-Video Generation` by Liu et al., `Domain Adaptation via Bidirectional Cross-Attention Transformer` by Wang et al., `Multimodal Transformer for Unaligned Multimodal Language Sequences`, `ViLBERT: Pretraining Task-Agnostic Visiolinguistic Representations for Vision-and-Language Tasks`, `Pano-AVQA: Grounded Audio-Visual Question Answering on 360◦ Videos`). Moreover the two years old MM-diffusion also employs dual cross attention for the same task.

**Questions:**

- What does the 1000 value correspond to between equations 5 and 6?
- It is common to employ temporally-aware positional encodings when interacting between video and audio modalities (that usually operate at different frame rates). Does the proposed model employ such design?

---

> ### Author Response · Authors · 2025-11-22
> **Official Response to Reviewer SK8e [1/2]**
>
> **Thank you very much for taking the time to review and for your support. We try our best to address your questions as follows.**
> ## Q1:  "Using different prompts for audio and video is clever but common sense."
> We respectfully clarify that our HVGC framework is not merely a "common sense" approach, but a targeted solution to specific T2SV challenges supported by rigorous empirical evidence.
>
> While using separate prompts might seem intuitive, it is **not standard practice in prior T2SV works** like CoDi and JavisDiT, which rely on a single shared prompt. We identify that shared prompts cause severe modal interference and propose disentangled text conditions as a specific remedy, validated by significant performance gains in Table 3. Meanwhile, high-quality academic video-audio datasets (e.g., AVSync15, VGGSS) currently **lack ready-to-use separated captions**. Our work fills this gap by providing a robust framework for generating separated captions and enriching benchmark datasets with this crucial data, supporting both training and evaluation in the separated paradigm.
>
> Disentangling text conditions for different modalities is technically non-trivial. Simply using an Audio LLM or a Single-Stage Omni-model often **leads to hallucinations due to the sparse information in audio**. Our core innovation is Visual Grounding: utilizing visual context to supervise audio captioning. This ensures the separate audio prompt is not just "different," but factually accurate and aligned with the video.
>
> To prove that our three-stage design is necessary (not just "additional complexity"), we conducted new ablation studies comparing HVGC against a Single-Stage Omni-model baseline and an HVGC variant w/o Stage 2.
>
> | Method | FVD ↓ | FAD ↓| CLIPSIM ↑ | CLAP ↑ | VA-IB ↑ | AVAlign ↑ |
> | :--- | :--- | :--- | :--- | :--- | :--- | :--- |
> | Single-Stage (Omini-model) | 1423.48 | 16.89 | 25.37 | 23.86 | 23.89 | 0.157 |
> | HVGC (Ours, No Stage2) | **908.81** | 15.48 | 28.34 | 25.48 | 23.32 | 0.198 |
> | HVGC (Ours, Full 3-Stage) | **908.81** | **14.90** | **28.34** | **28.37** | **25.36** | **0.211** |
>
> The results show that our full HVGC framework outperforms the simpler single-stage baseline. This confirms that our HVGC is essential for generating high-quality, low-hallucination prompts.
>
> ## Q2:  "The visually grounded audio captioning tasks lack a related works section."
>
> Thank you for this suggestion. We agree that adding a related works section will provide a better understanding of our method's contribution. We will add a "Visually-Grounded Audio Captioning" section to the related work in our revised version. Below is a summary of the key points we will discuss:
>
> Generating captions solely from audio is challenging due to the lower information density of sound compared to other modalities like speech or images [1]. Direct audio captioning methods [2, 3] often suffer from severe hallucinations and a lack of detail [4, 5]. For instance, as shown in Table 3, captions generated by a baseline Audio LLM fail to accurately capture specific sounds and their temporal relationships, significantly impacting the quality of the generated video. Recognizing the inherent correlation between audio and visual information, recent works have started leveraging visual cues to assist audio captioning [6, 7]. Our approach builds upon this trend but distinguishes itself by using a hierarchical approach that leverages video captions for layer-wise training, rather than directly using video features for training. This allows us to generate more accurate and detailed audio captions, as evidenced by our experimental results.

---

> > ### Author Response · Authors · 2025-11-22
> > **Official Response to Reviewer SK8e [2/2]**
> >
> > ## Q3: Lack of Citation for DCA
> > Thanks for your insightful comments. We have added citations to these works in our revised version. Below, we provide some discussions on the relationship between these works.
> >
> > While DCA may be used for foundational design, **demonstrating its potential in a new task holds significant academic value**. Just as the Transformer, proposed in 2017, took years to be widely adopted in vision understanding (ViT, 2020 [8]) and later in vision generation (DiT, 2022 [9]; Sora, 2024 [10]), DCA's application evolves. While DCA has addressed modal interaction in other tasks and unconditional generation in specific domains (MM-Diffusion), our experiments validate DCA's potential in the more general, open-domain T2SV task.
> >
> > Furthermore, our contribution is not simply designing or applying a DCA module, but rather identifying the optimal interaction mechanism for a dual-stream sounding video generation architecture. Given the novelty and growing interest in the T2SV task, prior dual-stream works lack detailed comparisons of interaction mechanisms. We address this gap by comprehensively comparing various modules, demonstrating that DCA offers the best overall performance. For a detailed analysis, please refer to Section 4.3.2.
> >
> > These methods also differ from ours fundamentally. Regarding MM-Diffusion (which we have already cited in our related work section), it is designed for unconditional generation. Meanwhile, MM-Diffusion employs a specialized MM-Block with Random-Shift based Multi-Modal Attention, which differs from our Dual-Cross Attention. The other cited papers similarly address single-modality generation or multi-modal understanding tasks, operating under different settings and objectives than our method.
> >
> > ## Q4： "What does the 1000 value correspond to between equations 5 and 6?"
> > The `1000` is a scaling factor. It is used to align the different timestep conventions of our pre-trained backbones. Our audio tower (Stable Audio) takes $t_a \in [0, 1]$ as timestep layer input, while our video tower (Wan) takes $t_v \in [0, 1000]$ as timestep layer input. Therefore, we define $t_v = 1000 \cdot t_a$ to synchronize them.
> >
> > ## Q5:  Question about positional encodings
> > This is an insightful point. We agree that a well-designed, temporally-aware positional encoding could further enhance video-audio interaction. However, given the simplified task mode (fixed duration and resolution) in our current work, we opted for a simpler design. Our BridgeDiT model relies on a standard 1D ROPE to mark the positions of video and audio tokens from the dual towers, and we found this sufficient to achieve synchronization.
> >
> > [1] Relja et al., "Look, listen and learn. 2017"
> >
> > [2] Knostantinos et al., "Automated audio captioning with recurrent neural network. 2017"
> >
> > [3] Yunfei Chu et al., "Qwen2-audio technical report. arXiv preprint arXiv:2407.10759, 2024."
> >
> > [4] Sung-Bin et al., "Avhbench: A cross-modal hallucination benchmark for audio-visual large language models 2024"
> >
> > [5] Nishimura et al., "On the audio hallucinations in large audiovideo language models. arXiv preprint arXiv:2401.09774, 2024"
> >
> > [6] Kim et al., "Avcap: Leveraging audio-visual features as text tokens for captioning, 2024."
> >
> > [7] Liu et al., "Visually-aware audio captioning with adaptive audio-visual attention 2023"
> >
> > [8] Alexey et al., "An Image is Worth 16x16 Words: Transformers for Image Recognition at Scale, 2020"
> >
> > [9] William et al., "Scalable Diffusion Models with Transformers, 2022"
> >
> > [10] Sora  https://openai.com/sora/

---

> > > ### Comment · Reviewer_SK8e · 2025-11-26
> > >
> > > I thank the authors for their clarifications, I will raise my score to 4. In light of the other discussions I am wondering whether DCA outperforming full attention is specific to the context of finetuning two pretrained towers or could be generalized to pre-training a joint audiovisual model.

---

> ### Author Response · Authors · 2025-11-26
> **Official Response to Reviewer SK8e**
>
> Dear Reviewer SK8e,
>
> Thank you for raising your score. We are happy to discuss with you on this fundamental question.   We have already explored this topic in detail in our `Official Response to Reviewer yKFY stage2[1/2]`. For your convenience, we will restate our analysis here.
>
> > I am wondering whether DCA outperforming full attention is specific to the context of finetuning two pretrained towers or could be generalised to pre-training a joint audiovisual model.
>
> We can explain our hypothesis from the perspective of the Manifold Assumption [1].
>
> Let's first consider the independent T2V and T2A generation. In this scenario, our video VAE and audio VAE are trained independently. We can therefore hypothesize that their latents, $z_v$ and $z_a$, exist on two distinct and uncorrelated latent manifolds, $M_v$ and $M_a$, respectively. Starting from a shared Gaussian prior $z \sim \mathcal{N}(0, I)$, the two generator branches ($G_\theta^V$ and $G_\theta^A$) learn to map this noise to their respective targets:
>
> $$
> G_\theta^V: z \to z_v, \quad \text{where } z_v \in M_v
> $$
>
> $$
> G_\theta^A: z \to z_a, \quad \text{where } z_a \in M_a
> $$
>
> When we aim for joint generation, our goal is to introduce an interaction module, or 'bridge' ($B_\theta^{AV}$), that connects these two branches. The full generative model is thus
>
> $$
> G_\text{model}= ( G_{\theta}^{A}, G_{\theta}^{V}, B_\theta^{AV} )
> $$
>
> Inside this bridge module, we must facilitate sufficient interaction. This is where the hypotheses for FA and DCA diverge.
>
> ### 1. Full-Attention (FA)
> From the Full-Attention perspective, we hypothesise that **FA works best by first "pulling"  the two distinct manifolds ($M_v, M_a$) into a single, unified joint manifold ($M_{joint}$)**. It attempts to use its QKV projection matrices to learn a shared representation *before* meaningful interaction can occur. This is a "merge-then-interact" approach:
>
> $$
> \text{FA path: } (L_v, L_a) \xrightarrow{\text{Project}} L_{joint} \xrightarrow{\text{Self-Attention}} L'_{joint} \to (L'_v, L'_a)
> $$
>
> This "manifold alignment" requires significant parameter capacity. In massive models like MMDiT [2] or SD3, which are trained with a vast number of parameters, FA is given enough power to successfully learn this complex $M_{joint}$ representation.
>
> ### 2. FA limitation
> However, our paradigm is different. As discussed in the paper, we operate under two crucial constraints:
>
> 1.  We use the dual-stream paradigm to leverage powerful pre-trained models.
> 2.  We must keep the backbone parameters **largely frozen** to retain this pre-trained knowledge.
>
> In this constrained scenario, the backbones themselves insist on operating within their original, separate manifolds ($M_v$ and $M_a$). FA is not given sufficient trainable capacity to "pull" these two powerful, separate streams into one. It struggles because its prerequisite (the ability to create a unified manifold) is not met.
>
> ### 3. Our DCA
> Our **Dual Cross-Attention (DCA)**, by contrast, is architecturally designed for this *exact* scenario. It does not need to align the manifolds. It operates as a lightweight "bridge" that facilitates direct, specialized interaction *between* the two independent streams:
>
> $$
> \text{DCA path: } (L_v, L_a) \xrightarrow{\text{Cross-Attn}} (L_v \leftarrow L_a, \ L_a \leftarrow L_v) \to (L'_v, L'_a)
> $$
>
>
> In this situation, DCA is tasked with learning the *mapping between* the manifolds ($M_v \leftrightarrow M_a$), which is a much more direct and well-defined task than FA's goal of creating an entirely new joint manifold ($M_{joint}$).
>
> ### 4. Q1 Summary
> In summary, because we chose the dual-stream, frozen-backbone paradigm (to leverage pre-training) and are thus limited in our parameter budget for interaction, FA cannot work effectively. **DCA provides a superior solution because it is specialised for interacting between separate manifolds.**
>
> Notably, we are not claiming that FA is universally inferior to DCA. We hypothesise that for a large-scale, AV-joint model trained from scratch with sufficient data and resources (which is hard), FA might be the superior choice. However, in the pre-trained, dual-tower paradigm, both our theory and experiments demonstrate that DCA is the more effective architecture.
>
> We will add this important discussion and analysis to the revised version of our paper.
>
> Please let us know if you have any further questions. We are happy to discuss. :)
>
> Best regards,
>
> [1] Back to Basics: Let Denoising Generative Models Denoise
>
> [2] Scaling Rectified Flow Transformers for High-Resolution Image Synthesis

---

### Author Response · Authors · 2025-11-23
**Official Comment by Authors**

In the revised paper,
1. We add a related work section on audio captioning tasks. Asked by Reviewer-SK8e.
2. We add more citations for dual-crossattention. Asked by Reviewer-SK8e.
3.  We add experimental setup details. Asked by Reviewer WTCT.
4. We add other fusion mechanisms beyond audiovisual alignment. Asked by Reviewer WTCT.
5. We add user study experiment with more human annotators. Asked by Reviewer WTCT.
6. We add details about spatial information in the audio. Asked by Reviewer K86T.
7. We add analysis of our DCA fusion. Asked by Reviewer yKFY.

Please see our detailed responses below each review for other questions and suggestions. We sincerely thank all reviewers for their thorough reviews and constructive feedback again.

---

> ### Author Response · Authors · 2025-11-24
> **A kind reminder for discussion**
>
> Dear reviewers,
>
> We express our sincere gratitude for your valuable comments, which we consider crucial for improving our work. This is a gentle reminder to reviewers to review our replies. We are eager to engage in an active discussion. Thank you again for your attention and consideration.

---

### Comment · Area_Chair_gsqL · 2025-11-26
**A Reminder on Your Crucial Role in the ICLR Discussion Period**

Dear Reviewers who haven't engaged with the rebuttal:

As the Area Chair, I would like to sincerely thank you for the time and expertise you have invested in writing your initial review. Your insights are invaluable to the decision-making process.

We are now entering the critical discussion and rebuttal phase. This is a collaborative process where authors have the opportunity to address your concerns and questions. Your active participation in this phase is essential to ensure we reach a fair and well-informed final decision.

I strongly encourage you to:

Engage with the Authors' Rebuttal: Please read the authors' response carefully and substantively.

Participate in the Discussion: Engage with the other reviewers on the forum. If the authors have clarified a point, please acknowledge it. If you have follow-up questions or remaining concerns, please voice them. Your dialogue with fellow reviewers is key to reaching a consensus.

Update Your Review (if necessary): Based on the discussion and rebuttal, you may feel the need to adjust your score or final recommendation. Please do so, as it reflects a more holistic view of the paper.

Your continued engagement ensures the integrity and quality of the ICLR conference. Thank you for your vital contribution to our community.

Best regards,

Area Chair gsqL, ICLR 2026

---

### Meta-Review · Area_Chair_sG7r · 2026-01-07

**Summary:**

Across reviewers, the main concerns that informed the decision were: (1) limited technical novelty, as disentangled prompting and cross-attention are largely incremental and context-dependent design choices rather than fundamentally new modeling ideas; (2) fairness of comparison; (3) unclear experimental setup; (4) more studies and analysis on DC; (5) limited user study; and (6) restricted scope of the task, as off-screen sounds and more realistic audio conditions were not considered.

**Reviewer Concerns:**

***Concerns substantially addressed by the rebuttal:***

Added ablation studies for the HVGC pipeline, including comparisons between multi-stage HVGC and single-stage Omni-model captioning.

Clarified the architectural distinction between Dual Cross-Attention (DCA) and Full Attention.

More fusion comparisons.

Demonstrated generalization by evaluating on additional backbones (e.g., SANA + AudioLDM).

Improved experimental clarity, including training protocols, data leakage checks, and BridgeDiT block.

Strengthened subjective evaluation by increasing the number of human annotators.

***Concerns still outstanding or only partially addressed:***

Reviewer WTCT remains concerned that comparisons against pretrained baselines are affected by caption and domain mismatch. The inability to retrain key baselines on HVGC captions limits how convincingly the results isolate the effectiveness of DCA.

It remains difficult to disentangle improvements due to DCA from those arising from HVGC caption quality and data filtering.

Several reviewers may still view the contributions as an incremental integration of known ideas rather than a source of new modeling insight.

The approach relies on strong filtering assumptions (e.g., focusing on on-screen sounds). Robustness to realistic, noisy, or misaligned audio-visual scenarios is not convincingly demonstrated.

**Reviewer Scores:**

Reviewer SK8e (initial score: 2 → updated to 4):
The reviewer acknowledged the clarifications and additional ablation studies provided in the rebuttal and raised their score accordingly. However, their central concern regarding limited technical novelty may remain unresolved.

Reviewer WTCT (initial score: 6):
This reviewer was generally positive and appreciated the added experiments, extended fusion comparisons, and the expanded user study. Nonetheless, they continued to express concerns about evaluation fairness, particularly the caption and domain mismatch between the proposed method and pretrained baselines. Following discussion, the reviewer explicitly stated they would maintain their score at 6.

Reviewer K86T (initial score: 2):
The authors provided additional backbone experiments, clarifications on data filtering, and more detailed methodological explanations. The reviewer engaged in the discussion but did not increase the rating.

Reviewer yKFY (initial score: 2 → updated to 4):
This reviewer raised substantial concerns regarding technical novelty and fairness of comparisons. The rebuttal addressed several concrete issues, including ablation studies, fusion mechanism comparisons, and data leakage checks, leading the reviewer to raise their score. However, the fundamental concern about limited technical novelty remains.

Although two reviewers indicated an increase in their ratings, both remained at a score of 4, and the only reviewer with a positive score continued to express substantive reservations. As a result, I do not believe that, even after full discussion, any reviewer would strongly champion this paper. Therefore, I defer to the reviewers’ overall assessments and recommend rejection.

---

### Decision · Program_Chairs · 2026-01-26

Reject